# Revealing an outward-facing open conformational state in a CLC Cl⁻/H⁺ exchange transporter

Chandra M Khantwal[1†], Sherwin J Abraham[1†], Wei Han[2,3,4,5], Tao Jiang[2,3,4,5], Tanmay S Chavan[1], Ricky C Cheng[1], Shelley M Elvington[1], Corey W Liu[6], Irimpan I Mathews[7], Richard A Stein[8], Hassane S Mchaourab[8], Emad Tajkhorshid[2,3,4,5]*, Merritt Maduke[1]*

[1]Department of Molecular and Cellular Physiology, Stanford University School of Medicine, Stanford, United States; [2]Department of Biochemistry, University of Illinois at Urbana-Champaign, Urbana, United States; [3]College of Medicine, University of Illinois at Urbana-Champaign, Urbana, United States; [4]Center for Biophysics and Computational Biology, University of Illinois at Urbana-Champaign, Urbana, United States; [5]Beckman Institute for Advanced Science and Technology, University of Illinois at Urbana-Champaign, Urbana, United States; [6]Stanford Magnetic Resonance Laboratory, Stanford University School of Medicine, Stanford, United States; [7]Stanford Synchrotron Radiation Lightsource, Stanford University, Menlo Park, United States; [8]Department of Molecular Physiology and Biophysics, Vanderbilt University, Nashville, United States

*For correspondence: emad@life.illinois.edu (ET); maduke@stanford.edu (MM)

[†]These authors contributed equally to this work

Competing interests: The authors declare that no competing interests exist.

**Abstract** CLC secondary active transporters exchange Cl⁻ for H⁺. Crystal structures have suggested that the conformational change from occluded to outward-facing states is unusually simple, involving only the rotation of a conserved glutamate ($Glu_{ex}$) upon its protonation. Using [19]F NMR, we show that as [H⁺] is increased to protonate $Glu_{ex}$ and enrich the outward-facing state, a residue ~20 Å away from $Glu_{ex}$, near the subunit interface, moves from buried to solvent-exposed. Consistent with functional relevance of this motion, constriction via inter-subunit cross-linking reduces transport. Molecular dynamics simulations indicate that the cross-link dampens extracellular gate-opening motions. In support of this model, mutations that decrease steric contact between Helix N (part of the extracellular gate) and Helix P (at the subunit interface) remove the inhibitory effect of the cross-link. Together, these results demonstrate the formation of a previously uncharacterized 'outward-facing open' state, and highlight the relevance of global structural changes in CLC function.

## Introduction

CLC transporters catalyze the exchange of Cl⁻ for H⁺ across cellular membranes (*Dutzler, 2007*; *Matulef and Maduke, 2007*; *Zifarelli and Pusch, 2007*; *Jentsch, 2008*; *Accardi and Picollo, 2010*; *Miller, 2015*; *Accardi, 2015*; *Jentsch, 2015*). In humans, they are critical to a wide variety of physiological processes and constitute therapeutic targets for treating diseases (*Jentsch, 2008*; *Zhao et al., 2009*; *Stauber et al., 2012*; *Stolting et al., 2014*; *Devuyst and Luciani, 2015*; *Pusch and Zifarelli, 2015*; *Zifarelli, 2015*). In bacteria and yeast, CLCs are virulence factors and therefore could serve as drug targets to protect against food poisoning and fungal infections (*Iyer et al., 2002*; *Zhu and Williamson, 2003*; *Canero and Roncero, 2008*).

**eLife digest** Cells have transporter proteins on their surface to carry molecules in and out of the cell. For example, the CLC family of transporters move two chloride ions in one direction at the same time as moving one hydrogen ion in the opposite direction.

To be able to move these ions in opposite directions, transporters have to cycle through a series of shapes in which the ions can only access alternate sides of the membrane. First, the transporter adopts an 'outward-facing' shape when the ions first bind to the transporter, then it switches into the 'occluded' shape to move the ions through the membrane. Finally, the transporter takes on the 'inward-facing' shape to release the ions on the other side of the membrane. However, structural studies of CLCs suggest that the structures of these proteins do not change much while they are moving ions, which suggests that they might work in a different way.

Khantwal, Abraham et al. have now used techniques called "nuclear magnetic resonance" and "double electron-electron resonance" to investigate how a CLC from a bacterium moves ions. The experiments suggest that when the transporter adopts the outward-facing shape, points on the protein known as Y419 and D417 shift their positions. Chemically linking two regions of the CLC prevented this movement and inhibited the transport of chloride ions across the membrane.

Khantwal, Abraham et al. then used a computer simulation to model how the protein changes shape in more detail. This model predicts that two regions of the transporter undergo major rearrangements resulting in a gate-opening motion that widens a passage to allow the chloride ions to bind to the protein. Khantwal, Abraham et al.'s findings will prompt future studies to reveal the other shapes and how CLCs transition between them.

From bacteria to humans, $Cl^-/H^+$ exchange by CLC transporters occurs with a strict stoichiometry of 2 $Cl^-$ for every $H^+$ (*Accardi and Miller, 2004*; *Picollo and Pusch, 2005*; *Scheel et al., 2005*; *Jayaram et al., 2011*; *Leisle et al., 2011*). To achieve this stoichiometric exchange, CLCs must follow an alternating access mechanism, in which bound substrate ions access either side of the membrane alternately, i.e., they cannot access both sides simultaneously (*Patlak, 1957*; *Jardetzky, 1966*; *Shilton, 2015*). The alternating access mechanism can only be realized by coupling of ion binding, translocation, and unbinding events to conformational changes in the transporter protein. Specifically, movement of ions between solution and the ion-binding sites of the transporter, as well as ion movement between binding sites, needs to be coupled to conformational changes between "outward-facing" (in which the external, but not internal, solution is accessible to ions), "occluded" (in which neither solution is accessible), and "inward-facing" (in which the internal, but not external, solution is accessible) states (*Forrest et al., 2011*; *Rudnick, 2013*).

In all other active transporters that have been structurally (or biophysically) characterized, the conformational changes governing the interconversion between these major functional states involve significant protein motions, including reorientation of helices or even entire domains (*Shi, 2013*; *Paulino et al., 2014*). For the CLC transporters, in contrast, it has been proposed that the transport mechanism may be fundamentally different and involve only localized side chain motions (*Feng et al., 2010*; *2012*). However, this proposed mechanism is based largely upon the observation that no large-scale CLC conformational change could be detected crystallographically. Given the strong constraining forces in a crystal environment, which often prevent the protein from populating all naturally accessible, functionally relevant conformational states (*Elvington and Maduke, 2008*; *Gonzalez-Gutierrez et al., 2012*; *2013*; *Kumar et al., 2014*), alternative approaches for detecting CLC conformational change during its function are strongly motivated.

CLC transporters are homodimers in which each subunit independently catalyzes $Cl^-/H^+$ antiport (exchange) (*Robertson et al., 2010*). There are two key $Cl^-$-binding sites within the protein lumen, known as $S_{cen}$ and $S_{ext}$. The central anion-binding site ($S_{cen}$) is stabilized by a positive electrostatic potential created by the N-termini of Helices F and N as well as by interactions with conserved Ser and Tyr residues, which physically occlude the anion from the intracellular side (*Figure 1A*). Using cross-linking as an alternative approach to crystallography, Basilio *et al.* demonstrated that the conserved Tyr contributes to an intracellular "gate" that opens to generate an inward-facing state

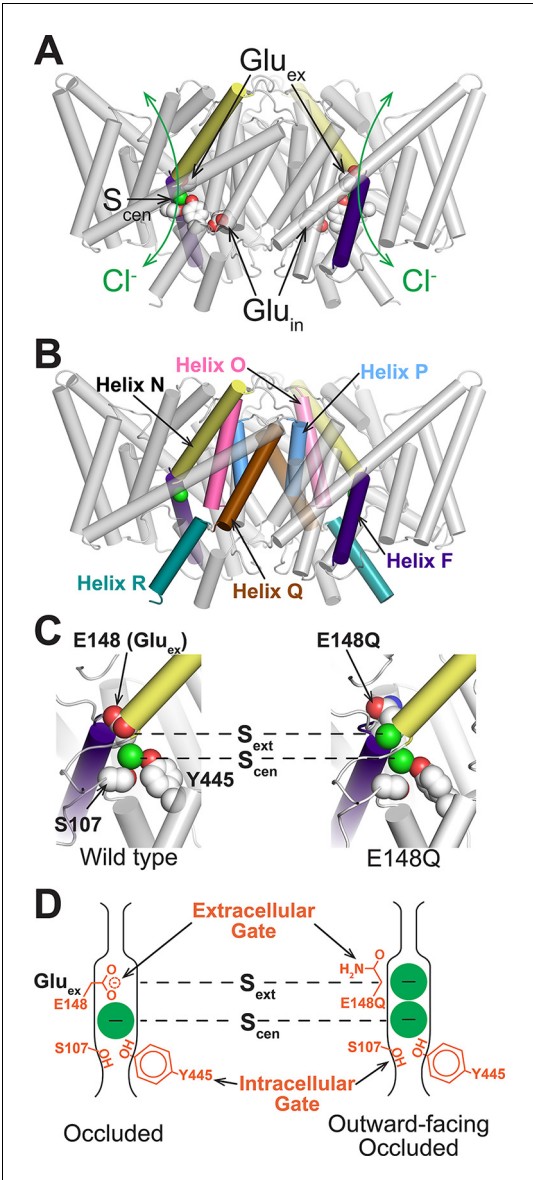

**Figure 1.** Structure of CLC transporters. (**A**) Structure of ClC-ec1 (pdb: 1OTS). The bound Cl⁻ (one in each identical subunit at site $S_{cen}$) is coordinated by conserved Ser and Tyr residues (shown as spacefilled). The N-termini of helices F and N (shown in purple and yellow respectively) point towards this site and provide a positive electrostatic environment for the anion. The H⁺-permeation pathways are delineated by two key residues, $Glu_{ex}$ and $Glu_{in}$. $Glu_{ex}$ also acts as a "gate" that blocks the Cl⁻-permeation pathway (green arrows) from the extracellular solution. (**B**) CLC structure highlighting helices discussed: F (purple), N (yellow), O (pink), P (blue), Q (brown), and R (aquamarine). (**C**) Close-up of the Cl⁻-binding region in WT (left) and E148Q (right) ClC-ec1, highlighting intracellular and extracellular gate residues S107, Y445, and E148 ($Glu_{ex}$). In the E148Q mutant (pdb: 1OTU), the Gln side chain, mimicking the protonated $Glu_{ex}$, swings away from the Cl⁻-permeation pathway and is replaced at $S_{ext}$ with a Cl⁻ ion. The structure of this mutant is otherwise indistinguishable from the WT structure. (**D**) Cartoon of the Cl⁻-binding region, illustrating the hypothesis that the E148Q structure represents an "outward-facing occluded" rather than an "outward-facing open" conformation.

The following figure supplement is available for figure 1:

**Figure supplement 1.** Comparison of CLC structures determined at high and low pH.

(*Basilio et al., 2014*). While this inward-facing state has not yet been structurally characterized in detail, the elegantly designed cross-linking studies demonstrated that movement of neighboring Helix O (*Figure 1B*) is required in conjunction with movement of the Tyr-gate residue.

At the extracellular side, a highly conserved glutamate residue, "$Glu_{ex}$", sits above the anion at $S_{cen}$ and blocks it from the extracellular solution (*Figure 1C*, *left panel*). Localized side-chain motions of this residue represent the sole differences distinguished in crystallographic studies of CLC transporters (*Dutzler et al., 2003*; *Feng et al., 2010*). In the structure of a mutant in which Gln is used as a proxy for the protonated $Glu_{ex}$, the side chain swings upwards and the site previously occupied by the side chain is occupied by an anion (*Figure 1C*, *right panel*). Thus, the structure of this mutant has been thought to represent an outward-facing (OF) CLC conformational state. However, in this structure the pathway to the extracellular solution is very narrow – too narrow to accommodate $Cl^-$ or other permeant ions (*Miloshevsky et al., 2010*; *Krivobokova et al., 2012*) – suggesting that additional conformational changes are required for the formation of the OF state in order for external anions to access the external anion-binding site ($S_{ext}$). We therefore hypothesize that the state identified in the E148Q crystal structure is an "outward-facing occluded" state and that a distinct "outward-facing open" state may exist to permit access of external $Cl^-$ to the $Glu_{ex}$-vacated $S_{ext}$ site (*Figure 1D*). Addressing this hypothesis is crucial to understanding the CLC transport mechanism and how it relates to those of canonical transporters.

Various experimental approaches have been used to evaluate whether CLC conformational changes beyond $Glu_{ex}$ are involved in the transition to an OF open state. Since the $pK_a$ of $Glu_{ex}$ is ~6 (*Picollo et al., 2012*), a change in pH from 7.5 to 5.0 will cause $Glu_{ex}$ to transition from mostly deprotonated to mostly protonated, and therefore from its position occupying $S_{ext}$ outward towards the extracellular solution. Such pH manipulations can therefore be used to enrich the OF state and probe for changes in protein conformation. Although crystallization at pH 4.6 failed to reveal any conformational change (*Figure 1—figure supplement 1*) (*Dutzler et al., 2002*), spectroscopic approaches have shown that $H^+$-dependent changes do occur outside the restraints of crystallization. Using environmentally sensitive fluorescent labels, Mindell and coworkers showed that Helix R, which lines the intracellular vestibule to the $Cl^-$-permeation pathway (*Figure 1B*), undergoes $H^+$-dependent conformational change during the transport cycle (*Bell et al., 2006*). Using site-specific NMR labeling schemes, our lab has identified $H^+$-dependent structural change at Helix R and also at the linker connecting Helices P and Q (P/Q linker) (*Figure 1B*) – a region ~20 Å distant from the $Cl^-$-permeation pathway (*Elvington et al., 2009*; *Abraham et al., 2015*). Clearly, CLCs undergo $H^+$-dependent conformational changes beyond those revealed by crystallography. The question remains whether and how these conformational changes are involved in regulating ion binding and translocation during $Cl^-/H^+$ transport.

Here, we study the conformational change in Helix P and the P/Q linker region (*Figure 1B*) in ClC-ec1, a well-studied prokaryotic CLC, using a combination of $^{19}F$ NMR, double electron-electron resonance spectroscopy, chemical cross-linking, crystallography, molecular dynamics (MD) simulations, and analysis of cross-linking in mutant transporters. Our results show that rearrangement of Helices N and P occurs to widen the extracellular vestibule and generate a previously uncharacterized "outward-facing open" CLC conformational state, thus establishing the involvement of structural changes beyond the rotation of $Glu_{ex}$.

## Results

### Sensitivity of $^{19}F$ spectra to the paramagnetic probe TEMPOL

The $^{19}F$ NMR nucleus is an advantageous reporter of conformational change because of its sensitivity to chemical environment, its small (non-perturbing) size, and the lack of endogenous $^{19}F$ in proteins (*Gerig, 1994*; *Danielson and Falke, 1996*; *Kitevski-LeBlanc and Prosser, 2012*). Using ClC-ec1, a prokaryotic CLC homolog that has served as a paradigm for the family (*Chen, 2005*; *Dutzler, 2007*; *Matulef and Maduke, 2007*; *Accardi, 2015*), we previously showed that we could replace native Tyr residues with $^{19}F$-Tyr and observe conformational changes reported by changes in $^{19}F$ chemical shift (*Elvington et al., 2009*). Our strategy to enrich the OF conformational state of ClC-ec1 involved lowering the pH of the solution from 7.5 to 4.5–5.0, as described above. Of the five buried Tyr residues in ClC-ec1, two reported [$H^+$]-dependent changes in chemical environment. The first, as expected,

was at Y445, which is within 6 Å of $Glu_{ex}$; the second, strikingly, was in a region ~20 Å away, at Y419 near the dimer interface (*Figure 2A*).

To better understand this conformational change, we performed an accessibility experiment, reasoning that global protein conformational changes often result in solvent exposure of previously buried regions. For this experiment, we exploited the sensitivity of $^{19}F$ relaxation rates (and hence spectral line widths) to the water-soluble paramagnetic probe TEMPOL (*Bernini et al., 2006*; *Venditti et al., 2008*). In this experimental setup, movement of a $^{19}F$-labeled residue from a buried to a solvent accessible location would be detected by line-broadening and peak attenuation. We first examined whether any of the five buried tyrosine residues in ClC-ec1 exhibits sensitivity to TEMPOL. In "BuriedOnly" ClC-ec1, a mutant in which all five buried Tyr residues have been labeled with $^{19}F$ (*Figure 2A*) effects of TEMPOL were observed at both low and high [$H^+$], with apparently greater sensitivity at high [$H^+$] (*Figure 2B*); however, because the $^{19}F$ resonances are overlapping, we were unable to unambiguously assign the observed changes specifically to effects on chemical shift or line-broadening of a particular resonance. Therefore, to clearly identify the residue(s) sensitive to TEMPOL, we generated ClC-ec1 constructs containing only one $^{19}F$-Tyr label per subunit (either Y445 or Y419, *Figure 2A*), replacing all other Tyr residues with Phe. Although the "Y445only" mutant was unstable and could not be further examined, the "Y419only" mutant (*Figure 3A*) was stable and showed robust, fully coupled $Cl^-/H^+$ exchange activity (*Figure 3—figure supplement 1*). The functionality of the Y419only mutant may seem startling, given that it involves mutating the highly conserved $Cl^-$-coordinating Tyr445 (*Figure 1C*) to Phe, but it is consistent with previous structural and functional studies demonstrating wild-type behavior of the Y445F mutant (*Accardi et al., 2006*; *Walden et al., 2007*).

## H⁺-dependent accessibility of Y419

Prior to investigating the effect of [$H^+$] on solvent accessibility of Y419 using TEMPOL, we acquired $^{19}F$ spectra for Y419 only as a function of pH. The $^{19}F$ NMR spectrum of Y419 only at pH 7.5 shows a single $^{19}F$ peak centered at 60 ppm (*Figure 3B*). This peak shifts upfield and splits into two peaks when [$H^+$] is increased, indicating that the $^{19}F$ nucleus has experienced a change in chemical environment. This result is consistent with our previous findings (*Elvington et al., 2009*) further supporting the notion that conformational changes occur in the vicinity of Y419 as increasing [$H^+$] promotes occupancy of the OF state in which $Glu_{ex}$ is protonated. The appearance of two peaks, at -61 and -63 ppm, indicates that the $^{19}F$ label on Y419 is experiencing two different environments. This could arise from two conformational states of ClC-ec1 or from a tyrosine ring flip that occurs slowly on the NMR timescale ($<<1000\ s^{-1}$) (*Weininger et al., 2014*). While this information is useful in identifying Y419 as being in a region involved in $H^+$-dependent conformational change, the lack of comprehensive theory for interpreting $^{19}F$ chemical shifts in terms of structure motivates additional studies to provide more details on the nature of the conformational change. To evaluate whether there might be a change in solvent accessibility of Y419, we examined the effect of TEMPOL on the $^{19}F$ spectra of Y419only. Because of the steep distance dependence of nuclear relaxation enhancements mediated by paramagnets like TEMPOL, significant line-broadening requires the paramagnetic center to approach the target nucleus within less than ~10 Å (*Teng and Bryant, 2006*). At pH 7.5, there is little sensitivity of the $^{19}F$-Y419 signal to 100 mM TEMPOL (*Figure 3C*) which is consistent with the largely buried position of Y419 in the crystallographically captured state of ClC-ec1, i.e., 12–13 Å from the protein surface (*Figure 3A*). In contrast, at pH 4.5, significant line-broadening is observed (*Figure 3C*), indicating exposure of Y419 to the bulk solution allowing a close approach, or direct contact, of the TEMPOL probe with the fluorine atom (*Esposito et al., 1992*; *Niccolai et al., 2001*). This $H^+$-dependent change in accessibility is reversible, as demonstrated by the reappearance of the Y419 signal when pH is returned to 7.5 from 4.5 in the presence of 100 mM TEMPOL (*Figure 3D*). The reproducibility of these experiments is shown in *Figure 3—figure supplement 2*.

## H⁺-independent accessibility of Y419 in channel-like ClC-ec1

The outer- and inner-gate residues of ClC-ec1 ($Glu_{ex}$ and Y445 respectively, *Figure 1*) can be replaced by smaller residues Ala, Ser, or Gly, to yield "channel-like" ClC-ec1 variants (*Jayaram et al., 2008*). This excavation of the gates yields a narrow water-filled conduit through the transmembrane domain, which allows rapid $Cl^-$ throughput and abolishes $H^+$ coupling. Thus, it

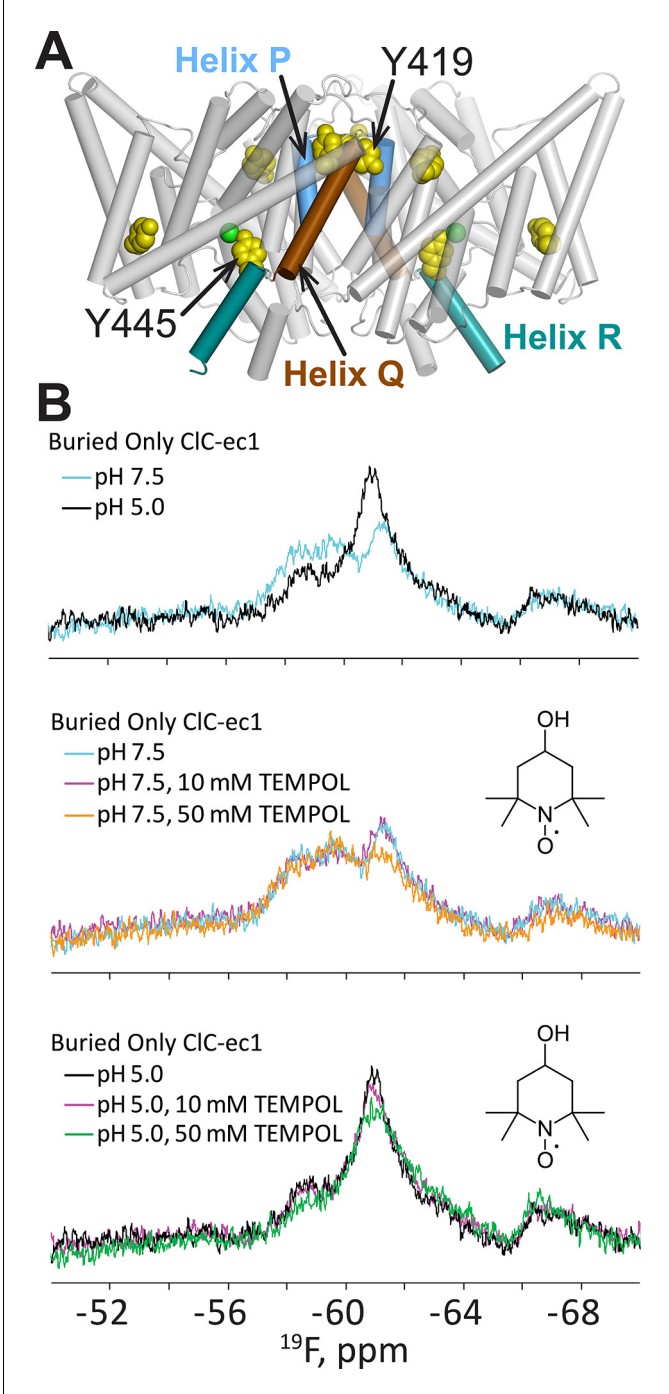

**Figure 2.** H[+]-dependent solvent accessibility of Tyr residues in ClC-ec1, detected by [19]F NMR. (**A**) "BuriedOnly" ClC-ec1, a mutant in which the five buried Tyr residues (spacefilled in yellow) have been labeled with [19]F. The seven solvent-exposed Tyr residues have been mutated to Phe. Residues Y445 (on Helix R, shown in aquamarine) and Y419 (linker between Helices P and Q, blue and brown respectively) were previously identified as undergoing H[+]-dependent changes in chemical shift (*Elvington et al., 2009*). (**B**) [19]F NMR spectra of BuriedOnly ClC-ec1. Top data panel: low pH was used to enrich the outward-facing conformational state. Changes in chemical shift reflect changes in chemical environment experienced by the [19]F nuclei. Middle data panel: spectral changes in response to addition of TEMPOL (inset) at pH 7.5. Bottom data panel: spectral changes in response to addition of TEMPOL at pH 5.0.

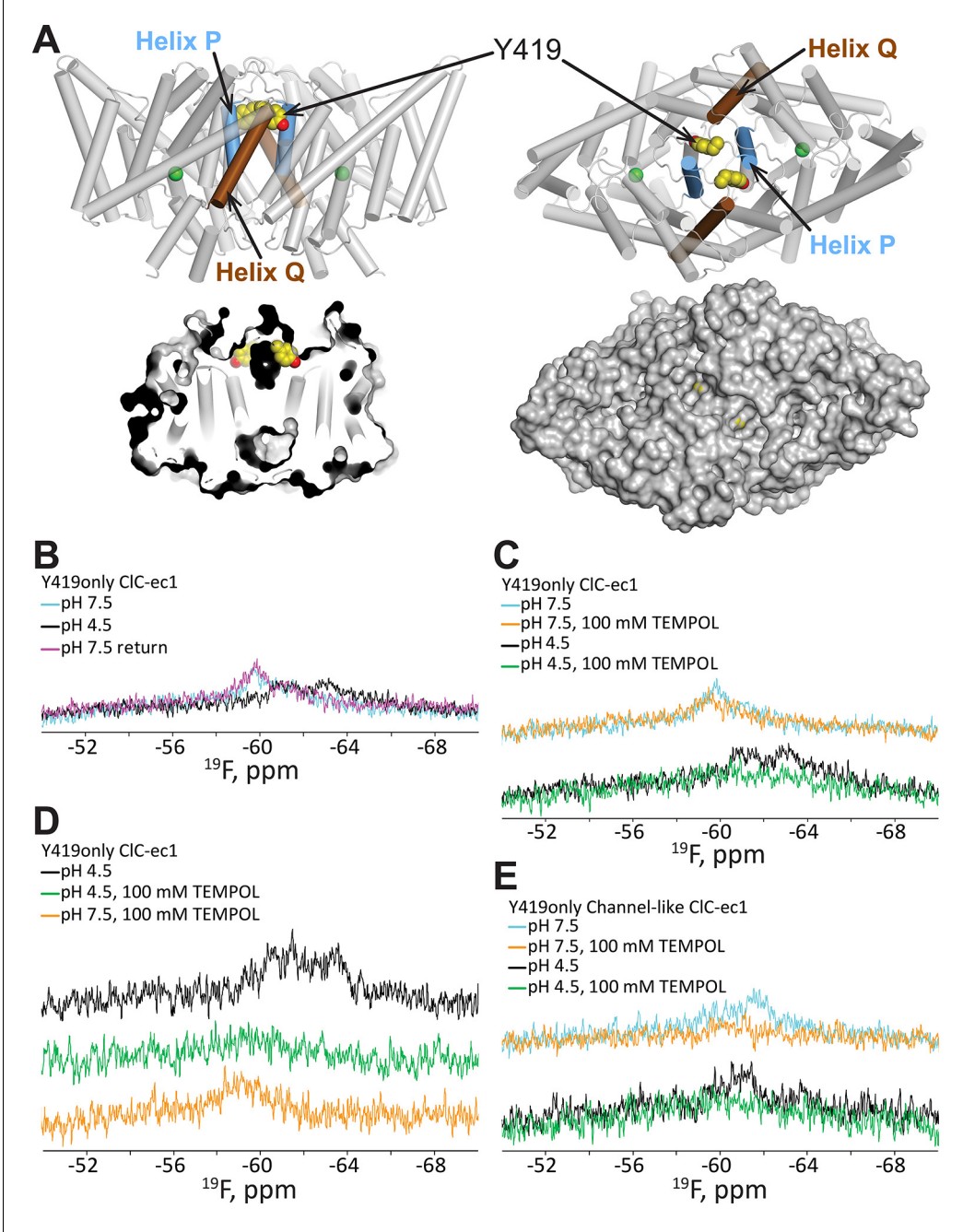

**Figure 3.** $^{19}$F NMR detects H$^+$-dependent solvent accessibility at Y419. (**A**) Y419only ClC-ec1. In this variant, all native Tyr residues except for Y419 have been mutated to Phe, so that only Y419 will carry a $^{19}$F label. Y419 is highlighted in the ClC-ec1 structure shown from the point of view of the membrane (left) and from the extracellular side (right). The lower panels illustrate that Y419 lies in a buried position (left: thin slice through the protein at Y419; right, surface representation viewed from the extracellular side. (**B**) $^{19}$F NMR spectra of Y419only. The prominent peak centered at -60 ppm shifts upfield (-61 and -63 ppm) when the pH is shifted from 7.5 to 4.5 to enrich the OF state. (**C**) Y419 becomes substantially more exposed to solvent at increased [H$^+$], as indicated by susceptibility to line-broadening by the water-soluble TEMPOL at pH 4.5 (bottom spectra, green vs black trace) compared to pH 7.5 (top spectra, orange vs cyan trace). (**D**) The change in the Y419 exposure to solvent is reversible, as revealed by return of the signal (to the expected chemical shift) when the pH is raised to 7.5 (bottom trace, orange). (**E**) Y419 in the channel-like ClC-ec1 background is accessible to TEMPOL at both pH 7.5 and 4.5.
The following figure supplements are available for figure 3:

*Figure 3 continued on next page*

*Figure 3 continued*

**Figure supplement 1.** Functional characterization of ClC-ec1 variants.
**Figure supplement 2.** Reproducibility of TEMPOL-NMR experiments.
**Figure supplement 3.** Overlay of WT and channel-like ClC-ec1 structures.

appears that the mechanism of Cl⁻ flux through these variants involves channel-like diffusion that is independent of substrate-dependent conformational change. Consistent with this picture, our previous $^{19}$F NMR data showed that E148A/Y445S ClC-ec1 (exhibiting the highest Cl⁻ permeability among the channel-like variants) does not undergo the substrate-dependent spectral changes observed in the coupled ClC-ec1 transporters (*Elvington et al., 2009*). In this channel-like background, Y419only exhibits a single NMR peak at ~-61 ppm, and this signal is sensitive to line-broadening by TEMPOL at both pH 4.5 and 7.5 (*Figure 3E*). The accessibility at pH 7.5 is surprising given that the crystal structure of channel-like ClC-ec1 variant E148A/Y445A superposes closely with WT (RMSD 0.52 Å) and indicates a buried position for Y419 (*Jayaram et al., 2008*) (*Figure 3—figure supplement 3*). While in these studies we used variant E148A/Y445S, which has not been crystallized, the two variants are functionally similar (*Jayaram et al., 2008*). In channel-like E148A/Y445S, the accessibility of Y419 to TEMPOL indicates that the channel-like ClC-ec1 variant adopts a conformation in solution different from that observed in the crystal structure and similar to the conformation adopted by WT at low pH.

## Cross-linking at Helix P slows transport

We investigated the functional relevance of the conformational change detected at Y419 by introducing cysteines into this region and examining the effects of inter-subunit cross-linking. We reasoned that this cross-linking would restrict the conformational changes responsible for the increased solvent accessibility of Y419 at low pH, and, if these conformational changes are functionally important, cross-linking should also reduce the efficiency of Cl⁻/H⁺ transport. In the X-ray crystal structure, the Cα-Cα distance between the two Y419 residues (one in each subunit) is 8.8 Å, within striking range for potential disulfide bond formation. We found that Y419C forms spontaneous inter-subunit cross-links and, to our surprise, that these cross-links have no detectable effect on function (*Figure 4—figure supplement 1*). Since Y419 lies in the middle of a loop (the P/Q linker), we reasoned that loop flexibility may thwart the intended restriction of motion by the disulfide cross-link. To test this possibility, we examined cross-linking at D417, the residue immediately following Helix P, which also has an inter-subunit Cα-Cα distance of 8.8 Å (*Figure 4A,B*). Like Y419C, D417C forms spontaneous inter-subunit disulfide cross-links, with ~50% of the protein migrating as a dimer on non-reducing SDS-PAGE (*Figure 4C*, top panel). To determine the effect of the cross-link on function, we purified D417C transporters under reducing conditions, thereby obtaining a sample in which the majority (>90%) of the protein was not cross-linked, and then induced varying amounts of cross-link by titrating with copper-phenanthroline (CuP) (*Figure 4C*, bottom panel). We assessed the functional effect of cross-linking using a Cl⁻ efflux assay (*Walden et al., 2007*). These assays show that cross-linking at 417C correlates directly with a decrease in transport activity (*Figure 4D*), with Cl⁻ and H⁺ transport inhibited in parallel (*Figure 4—figure supplement 2*). Controls showing the lack of effect of CuP on WT and cysteine-less transporters are shown in *Figure 4—figure supplement 3*. Linear extrapolation to 100% cross-linking is summarized for all D417C variants in *Table 1*.

Since our NMR results indicated that the Helix P-Q region of channel-like ClC-ec1 adopts a conformation similar to that of WT at low pH (*Figure 3*), we hypothesized that the conformational changes underlying the increased solvent accessibility of Y419 may also move the two D417 side chains out of the range for inter-subunit cross-linking. To test this hypothesis, we generated the D417C mutant in the channel-like background and evaluated its sensitivity to cross-linking. Consistent with our hypothesis, D417C in channel-like background does not form spontaneous cross-links and is only minimally cross-linked even when treated with up to 100 μM CuP (*Figure 4E*, top panel). Because this limited cross-linking of D417C/channel-like ClC-ec1 could be due to non-availability of the cysteines (due to oxidation) rather than lack of structural proximity of the two cysteine residues,

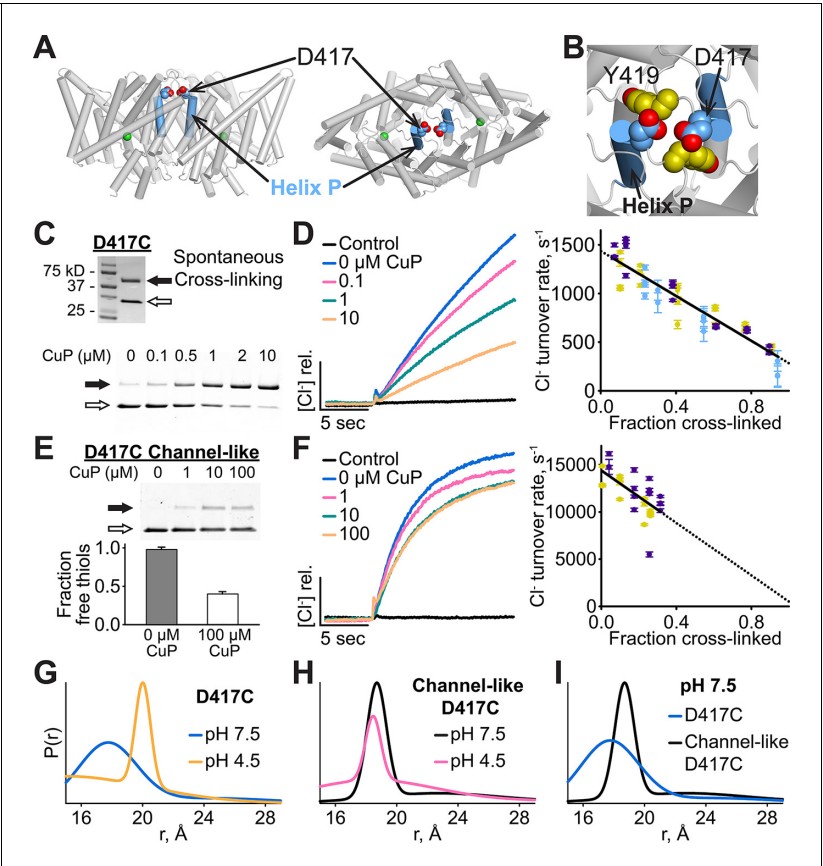

**Figure 4.** Cross-linking and H⁺-dependent conformational change at D417C. (**A**) ClC-ec1 with D417 side chain shown spacefilled, viewed from the membrane (left) and from the extracellular side (right). (**B**) Close-up view showing D417 and Y419 side chains. (**C**) Detection of inter-subunit disulfide cross-links by non-reducing SDS-PAGE. When the D417C transporters were purified under standard (non-reducing) conditions, inter-subunit cross-links formed spontaneously, with ~50% of the protein migrating as a dimer (top gel, solid arrow) and ~50% as a monomer (open arrow). By purifying the transporters under reducing conditions, the amount of cross-linking could be reduced to <10%, and then titrated (up to ~95%) with addition of increasing amounts of CuP (bottom panel). (**D**) Effect of cross-linking on D417C activity. *Left*: Representative data traces showing Cl⁻-transport activity of D417C. *Right*: Summary data showing Cl⁻-transport activity as a function of disulfide cross-linking, which was determined by quantifying the relative intensities in the monomer and dimer bands detected by SDS-PAGE (as shown in panel C). Each data point represents one flux-assay measurement. Error bars (most are smaller than the symbols) indicate the uncertainty in curve-fitting to the primary data (transporter flux and background leak measured in control liposomes). Data are from three separate D417C ClC-ec1 preparations, as depicted by three colors (purple, yellow, and blue). (**E**) D417C/channel-like is resistant to CuP-induced cross-linking. The bottom panel shows results from thiol quantification assays before and after treatment with 100 μM CuP. (**F**) Effect of cross-linking on activity of D417C/channel-like ClC-ec1. *Left:* Representative data traces. *Right*: Summary data, as in panel D. Yellow and purple indicate data from two separate D417C/channel-like ClC-ec1 preparations. (**G**) DEER distance distributions reveal a pH-dependent increase in inter-subunit distance at D417C. (**H**) D417C/channel-like does not exhibit the pH-dependent change observed with D417C/WT. (**I**) Comparison of WT and channel-like D417C at pH 7.5.

The following figure supplements are available for figure 4:

**Figure supplement 1.** Cross-linking of Y419C ClC-ec1.

**Figure supplement 2.** Cross-linking at D417C inhibits Cl⁻ and H⁺ transport in parallel.

**Figure supplement 3.** Control experiments on WT and cysteine-less ClC-ec1.

**Figure supplement 4.** CuP-treated D417C proteins run as dimers on size exclusion chromatography.

**Figure supplement 5.** Functional-, CW-EPR, and DEER data analysis for spin-labeled D417C variants.

we used a spectrophotometric assay to quantify the free thiols. Immediately after purification and before CuP treatment, essentially all of the 417C residues are available as free thiols (*Figure 4E*, bottom panel). Therefore, the deficiency in cross-linking of 417C in the channel-like background compared to 417C in the WT background is not due to unavailability of the free thiols but rather because of a change in proximity of the two cysteines. After treatment with 100 µM CuP, 40% of the cysteines are available as free thiols (*Figure 4E*). Since only ~25% had been cross-linked, this result indicates that ~35% became oxidized to other (non-disulfide) species. Thus, oxidation to non-disulfide species competes with disulfide bond formation and therefore thwarts any attempt to increase the extent of cross-linking beyond ~25% with longer CuP treatments. To rule out the possibility that the crosslinking might be due to inter-dimer (rather than inter-subunit) disulfide bond formation, we examined D417C proteins on a Superdex 200 gel filtration column both before and after treatment with CuP and found that they ran as dimers (and not tetramers, as would occur in the case of inter-dimer cross-linking) (*Figure 4—figure supplement 4*). To the degree it can be cross-linked, D417C in channel-like ClC-ec1 is inhibited similarly to D417C in WT (*Figure 4F*). Thus, movement of Helix P away from the position observed in the crystal structures is necessary for maximal activity in both transporter and channel-like ClC-ec1.

To evaluate whether this movement of D417C/Helix P is $H^+$-dependent (as is the movement detected by $^{19}F$ NMR, *Figure 3*), we labeled D417C ClC-ec1 with the nitroxide spin label MTSSL (1-Oxyl-2,2,5,5,-tetramethylpyrroline-3-methyl methanethio-sulfonate) and used double electron-electron resonance (DEER) spectroscopy (*Jeschke, 2012*) to deduce distance changes as a function of pH. At pH 7.5, the distribution is dominated by a single peak at a distance shorter than 20 Å (*Figure 4G*). Lowering the pH to 4.5 induces a shift to a peak at 20 Å (*Figure 4G*). The channel-like D417C-MTSSL exhibits an altered distance distribution profile compared to the WT background (*Figure 4H*), suggesting that the protein adopts a different conformation. Notably, the D417C-D417C distance in channel-like is increased relative to that observed in the WT background at pH 7.5 (*Figure 4I*), consistent with the resistance of the channel-like protein to cross-linking (*Figure 4E*). A decrease in pH does not shift the distance distribution as observed in the WT background, consistent with the loss of pH dependence in $^{19}F$ NMR experiments on channel-like ClC-ec1 (*Figure 3E*, (*Elvington et al., 2009*)). Together, these results link the conformational changes observed via NMR to those prevented by the cross-link.

Crystallization of cross-linked D417C (WT background) confirms that the cross-link has trapped the conformation seen in the crystal structures and not some other (non-native) conformation. Our crystal structure, determined at 3.15 Å resolution, reveals a backbone that superimposes on WT ClC-ec1 with a Cα RMSD of 0.57 Å (*Figure 5A*, *Table 2*). Extra density connecting the 417C residues confirms the formation of an inter-subunit disulfide bridge (*Figure 5B*). The regions around both the Cl⁻ and the H⁺ permeation pathways are intact and not notably distinguishable from WT (*Figure 5C*). To confirm the integrity of the Cl⁻-permeation pathway in cross-linked D417C ClC-ec1, we directly measured Cl⁻-binding affinity using isothermal titration calorimetry (ITC) (*Picollo et al.*,

**Table 1.** D417C activity extrapolated to 0 and 100% cross-link.

| D417C variant | Turnover at 0% cross-link (s⁻¹) | Turnover at 100% cross-link (s⁻¹) |
|---|---|---|
| WT | 1440 ± 70 | 280 ± 70 |
| E148A/Y445S (channel-like) | 14400 ± 1300 | 520 ± 5400 |
| E148A | 260 ± 20 | 76 ± 26 |
| Y445S | 850 ± 70 | -20 ± 105 |
| A404L | 140 ± 20 | 54 ± 17 |
| L361A (Helix N) | 360 ± 30 | 240 ± 30 |
| F357A (Helix N) | 84 ± 9 | 119 ± 8 |

Values for turnover at 0 and 100% D417C cross-link were estimated from extrapolation of fits to data in *Figures 4*, *7* and *11*. The uncertainties report the 95%confidence interval in the extrapolated values.

*2009*). Both in the absence and presence of cross-link, D417C binds Cl⁻ robustly, with an affinity somewhat stronger than observed with WT ($K_d$ ~0.1–0.2 mM vs 0.6 mM) (*Figure 5D,E*).

## Helix P cross-link specifically affects the Cl⁻-permeation pathway

The inhibition of channel-like ClC-ec1 activity by the D417C cross-link suggests that inhibition occurs via an effect on the Cl⁻-permeation pathway, given that channel-like ClC-ec1 transports only Cl⁻ and not H⁺. But since this conclusion is based on experiments with the atypical channel-like ClC-ec1 – with high transport rate and a continuous water passageway connecting the two sides of the membrane – we sought to strengthen the conclusion by examining uncoupled transporters that display typical transport rates and lack a continuous passageway: (1) E148A, which lacks the critical Glu$_{ex}$ residue that acts both as an extracellular gate for Cl⁻ and as a transfer-point for H⁺ permeation (*Figure 1*), and (2) Y445S, which is mutated at the intracellular gate (*Basilio et al., 2014*) (*Figures 1, 2*). The E148A (Glu$_{ex}$) mutant is similar to channel-like ClC-ec1 in that it transports only Cl⁻; however, it has a much lower turnover rate, comparable to WT (*Accardi and Miller, 2004*; *Jayaram et al., 2008*). This slow turnover suggests that despite being uncoupled E148A still depends on conformational changes to catalyze transport, a view supported by both ¹⁹F NMR and fluorescence-based experiments, which detect H⁺-dependent conformational change in this mutant (*Bell et al., 2006*; *Elvington et al., 2009*). In contrast to D417C/channel-like ClC-ec1, we found that D417C in the E148A mutant background is readily cross-linked by CuP (*Figure 6A*). Therefore, uncoupling through E148A alone does not alter the protein conformations sampled in solution as substantially as observed with channel-like (E148A/Y445S). The turnover rate of un-crosslinked D417C/E148A is quite low – as low, in fact, as the extrapolated value for the turnover of fully cross-linked D417C (*Table 1*). Nevertheless, cross-linking of D417C/E148A is associated with significant inhibition of activity (*Figure 6B*, *Table 1*). This result is consistent with the conclusion that inhibition occurs via an effect on the Cl⁻-permeation pathway.

The second uncoupled transporter examined, the inner-gate mutant Y445S, differs from the Glu$_{ex}$ mutant in that it is only partially uncoupled, with a Cl⁻/H⁺ stoichiometry of ~39:1 instead of the 2:1 stoichiometry observed with WT transporters (*Walden et al., 2007*). The double mutant D417C/Y445S transports Cl⁻ at ~850 s⁻¹ (*Figure 6C*) and H⁺ at ~20 s⁻¹ (*Figure 6—figure supplement 1*) yielding a Cl⁻/H⁺ stoichiometry of ~43, similar to that of the Y445S single mutant (*Walden et al., 2007*). Cross-linking of D417C/Y445S proceeds to ~70% and inhibits Cl⁻ transport by ~60% (*Figure 6D*), with extrapolation to 100% cross-linking yielding a turnover of ~0 (± 100 s⁻¹) (*Table 1*). For H⁺ turnover, it is difficult to judge whether there is a significant effect of the cross-link (*Figure 6—figure supplement 1*). Given the uncertainty in measuring such low H⁺ fluxes (~20 s⁻¹), it may be that H⁺ is inhibited to the same extent as Cl⁻, to a lesser degree, or not at all. In the latter cases, the cross-link would in effect "rescue" ClC-ec1 coupling; such rescue could arise from an increase in Cl⁻ occupancy at S$_{cen}$, which is known to facilitate H⁺ coupling (*Accardi et al., 2006*; *Nguitragool and Miller, 2006*; *Han et al., 2014*). However, we cannot distinguish these possibilities within the uncertainty of our measurements.

The results with channel-like and uncoupled transporters (*Figures 4, 6*) support the conclusion that the D417C cross-link inhibits the Cl⁻ branch of the Cl⁻/H⁺ transport mechanism but do not rule out an effect on H⁺ transport. As an approach to examine the effect of the cross-link on H⁺ transport, we used MD simulations to examine water entry into the hydrophobic region between Glu$_{in}$ and Glu$_{ex}$, which is essential to connect these two major H⁺-binding sites and thus support H⁺ transport (*Kuang et al., 2007*; *Wang and Voth, 2009*; *Cheng and Coalson, 2012*; *Lim et al., 2012*; *Han et al., 2014*). Water entry occurs via a narrow portal on the cytoplasmic side of the protein, lined by Glu$_{in}$ together with E202 and A404 (*Lim et al., 2012*; *Han et al., 2014*). Previously, we showed that constricting this portal by introducing large side chains at position 404 inhibits water entry detected computationally and H⁺ transport detected experimentally (*Han et al., 2014*). Since A404 is on the intracellular end of Helix P (*Figure 7A*), restricting movement of this helix via the D417C cross-link might restrict water entry. To determine whether the D417C cross-link affects water entry, we compared the number of water molecules entering the central hydrophobic region during the simulation of cross-linked D417C compared to WT. In contrast to the A404L mutation, which greatly reduces water permeation through the portal (*Han et al., 2014*), the D417C cross-link has no effect on water entry (*Figure 7B*). This result suggests that the cross-link reduces ClC-ec1

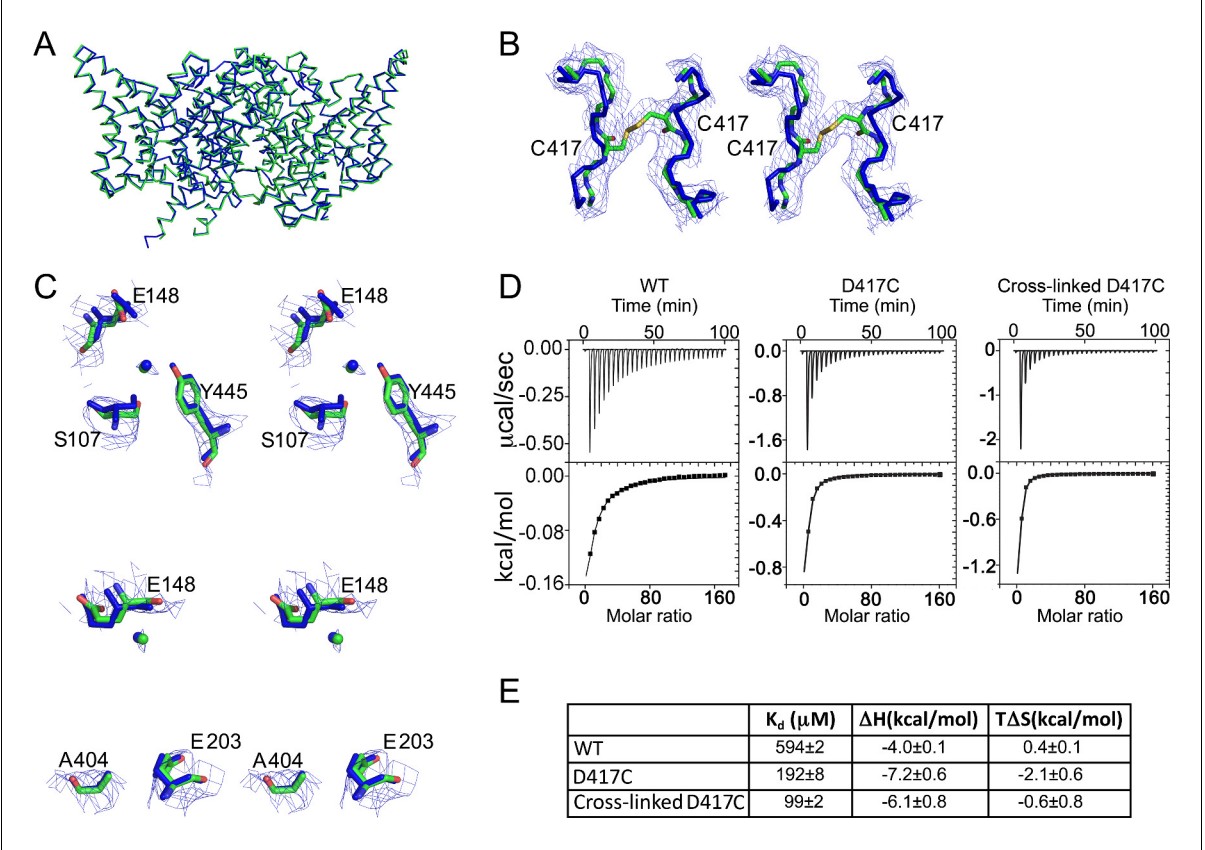

**Figure 5.** Structural integrity of cross-linked D417C. (**A**) The cross-linked D417C backbone (PDB 5HD8, green) superposes with WT (PDB 1OTS, blue) (RMSD 0.57 Å for 862 Cα atoms). (**B**) Extra density between residues 417 on the two subunits was modeled as a disulfide bridge, shown in stereo. (**C**) Close up stereo view of key residues around the Cl⁻ (upper panel) and H⁺ (lower panel) permeation pathways. In the upper panel, the residues shown (E148, S107, and Y445) are the same as those depicted in *Figure 1A*. In the lower panel, also shown are E203, the internal H⁺-transfer site (*Accardi et al., 2005*) and A404, a residue lining the portal for H⁺ entry from the intracellular solution (*Han et al., 2014*). Cl⁻ modeled in the central binding site is depicted as green and blue spheres. $2F_0$-$F_c$ maps are contoured at 1σ. (**D**) ITC experiments show Cl⁻ binding to WT, D417C, and D417C cross-linked with 100 µM CuP. Top panels: heat liberated when 20 mM KCl is titrated into the ITC cell containing 25–50 µM protein (WT, 25 µM; D417C, 50 µM; D417C+CuP, 30 µM). (**E**) Summary data for ITC experiments, ± SEM. WT, n=3 from two separate protein preparations; D417C, n=4 from four separate preparations; cross-linked D417C n=4 from three separate protein preparations.

transport predominantly via an effect on the Cl⁻-permeation pathway rather than on the H⁺-permeation pathway.

## Potential gate-opening motions in WT and cross-linked ClC-ec1

Our experimental results suggest that there could be functionally important motions of ClC-ec1 that open the Cl⁻-transport pathway and are impeded by the D417C cross-link. To investigate the molecular basis of such motions, extensive MD simulations were conducted either in the absence or in the presence of the D417C cross-link. Note that even the hundreds of nanoseconds of simulations performed here can probe mainly conformational fluctuations of ClC-ec1 near its reference conformation (in this case, the crystal structure), which did not permit direct observation of the opening of the gates. Nevertheless, the sampled dynamics and fluctuations can provide information that can be used to derive collective motions, which are often functionally relevant (*Bahar et al., 2010*). Collective motions are defined as those involving concerted movements of a large number of atoms distributed throughout the protein, and are therefore distinguished from localized conformational changes. A series of collective motions of a protein can be obtained in general by decomposing the fluctuations of a protein sampled through MD simulations, e.g., through principal component analysis, or by analyzing normal modes of the protein that underlie protein motions (*Bahar et al., 2010*;

**Table 2.** Data collection and refinement statistics[a].

| Data collection | |
| --- | --- |
| Space group | C121 |
| Unit cell dimensions | |
| a, b, c (Å) | 231.7, 96.1, 170.0 |
| α, β, γ (°) | 90, 132, 90 |
| Resolution range (Å) | 39.2–3.15 (3.23-3.15) |
| Completeness (%) | 90.2 (80.6) |
| $R_{merge}$ (%) | 7.7 (70.9) |
| I/σ (I) | 14.8 (1.7) |
| Redundancy | 3.6 (2.1) |
| **Refinement statistics** | |
| Resolution limit (Å) | 39.2–3.15 |
| No. of reflections | 41,839 |
| $R_{work}$/$R_{free}$ (%) | 20.5/25.7 |
| **Number of atoms** | |
| Protein | 13,064 |
| Ligand/ion | 4 |
| **B-factors** | |
| Protein | 69.8 |
| Ions | 118.4 |
| **r.m.s deviations** | |
| Bond lengths (Å) | 0.007 |
| Bond angles (°) | 1.139 |

[a]Values in parentheses are for the highest-resolution shell. Data were collected from a single crystal.

*Gur et al., 2013*). Collective motions can further be used to probe how larger-magnitude conformational change along the identified displacement vectors (modes) might involve crucial, functionally-relevant protein motions, such as opening-closing movements of enzymatic active sites, ligand-binding sites on receptors and channel pores (*Tai et al., 2001*; *Lou and Cukier, 2006*; *Shrivastava and Bahar, 2006*; *Liu et al., 2008*; *Jiang et al., 2011*; *Isin et al., 2012*; *Peters and de Groot, 2012*; *Fan et al., 2013*; *Yao et al., 2013*). For example, collective motions obtained from normal mode analysis (NMA) were used to project opening movements of potassium-channel pores (*Shrivastava and Bahar, 2006*), and these predicted movements are consistent with those seen in single-molecule and X-ray crystallographic experiments (*Shimizu et al., 2008*; *Alam and Jiang, 2009*). In this study, we identified collective motions in ClC-ec1 using principal component analysis (PCA) of the equilibrium MD simulations (see Methods), which in general identifies similar collective motions to those derived from NMA (*Leo-Macias et al., 2005*; *Yang et al., 2008*; *Skjaerven et al., 2011*). We then introduced deformations in the reference protein structure along each of the top 20 collective motions identified in our analysis (~75% of the motions observed in the equilibrium MD simulation). We then examined whether increasing the amplitude of these collective motions (which overcomes timescale limitations of the simulation) confer conformational change to the Cl⁻-transport pathway. We specifically examined regions around the extracellular and intracellular gates to the Cl⁻-transport pathway, where motions may lead to opening of either gate (which are both closed in the reference protein structure).

The extracellular gate is formed by the juxtaposition of Helix F (which contains Glu$_{ex}$) and Helix N (*Figure 1B*). To scrutinize opening of this gate, we examined Cα distance changes (Δr) between several residue pairs on these helices: I356-G149, F357-E148, and A358-R147 (*Figure 8A,B*). A search over the 20 dominant collective motions obtained through the PCA of the entire WT MD simulation

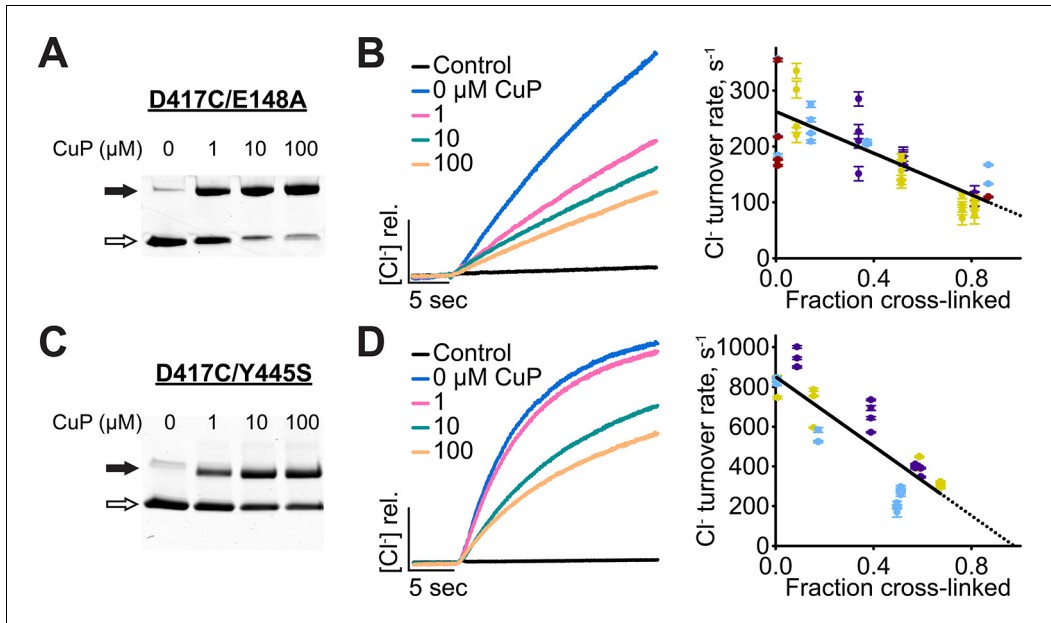

**Figure 6.** Cross-linking D417C in uncoupled transporter backgrounds. (**A**) D417C/E148A – detection of inter-subunit disulfide cross-links by non-reducing SDS-PAGE. (**B**) Effect of cross-linking on activity of D417C/E148A. *Left:* Representative data traces showing Cl⁻-transport activity. *Right*: Summary data showing Cl⁻-transport activity as a function of disulfide cross-linking. Each data point represents one flux-assay measurement, with error bars indicating the uncertainty in curve-fitting to the primary data. Purple, yellow, blue, and dark red each represent data from a separate protein preparation. (**C**) D417C/Y445S – detection of inter-subunit disulfide cross-links. (**D**) Effect of cross-linking on activity of D417C/Y445S, as in panel B. Data are from three separate protein preparations (indicated in purple, yellow, and blue).
The following figure supplement is available for figure 6:

**Figure supplement 1.** H⁺ turnover of D417C/Y445S.

revealed that deformations along some of the collective motions increase the distances between these pairs by >1.5 Å and thus are coupled to gate opening. To conduct a statistical analysis of these motions, we divided the entire simulation trajectory into six blocks and determined the number of times such collective motions occur in each block (*Figure 8C*, blue bars). An identical analysis performed on the MD simulation trajectory obtained from the cross-linked D417C mutant revealed that the motions that open the extracellular gate are dampened due to the cross-linking (P=0.002 – 0.008) (*Figure 8C*, orange bars). The intracellular gate is formed by two key residues S107 and Y445 (*Walden et al., 2007*; *Accardi and Picollo, 2010*; *Basilio et al., 2014*) (*Figure 2A*). To scrutinize opening of this intracellular gate, we examined distance changes between these two residues as a result of collective motions. As with the extracellular gate, we observed some collective motions that lead to distance changes ($\Delta r$) of > 1.5 Å. Unlike the extracellular gate, however, the cross-link at residue 417 does not significantly dampen the distance changes around the intracellular gate (P=0.338) (*Figure 8D*).

## Collective motions in channel-like ClC-ec1

The comparison of dominant gate-opening motions between WT and cross-linked forms described above suggests that the cross-link at residue 417 likely cripples the opening of the extracellular gate, thereby slowing Cl⁻ transport. However, along this line of reasoning, one must reconcile why the E148A mutants, in which the extracellular gate has ostensibly been removed, are inhibited when the cross-link is introduced. To address this question, we first investigated the bottleneck for Cl⁻ transport in ClC-ec1 based on the crystal structures. The radius profile of the ClC-ec1 Cl⁻ transport tunnel, calculated using the program HOLE (*Smart et al., 1996*), shows an extracellular bottleneck

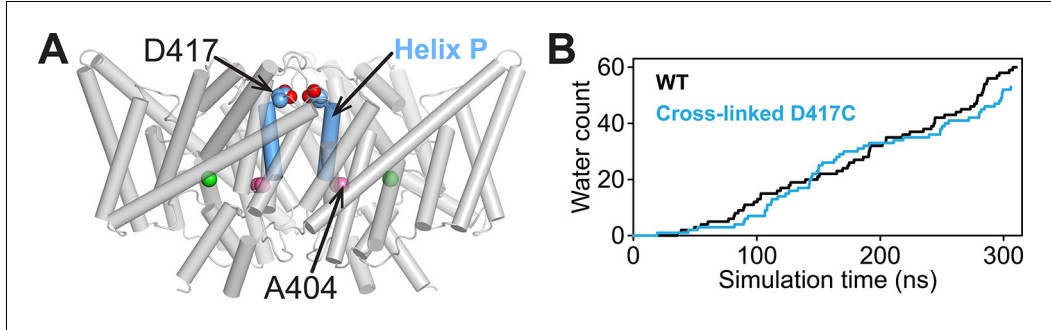

**Figure 7.** Computational analysis of water entry through the portal lined by A404 (Helix P). (**A**) ClC-ec1 structure highlighting the location of the A404 "portal" residue at Helix P. (**B**) The D417C cross-link does not affect water entry into the pathway connecting Glu$_{in}$ and Glu$_{ex}$. The aggregate number of water molecules entering the region between the two residues was determined as described previously (**Han et al., 2014**) and compared for wild-type (WT) and cross-linked mutant (D417C) over the same timescales.

with a minimum radius of ~0.2 Å (**Figure 9**). Interestingly, the calculated radius profile for both the E148A mutant (lacking Glu$_{ex}$) and the channel-like variant E148A/Y445A also reveal extracellular bottlenecks. (E148A/Y445A was evaluated rather than the E148A/Y445S construct used here because this is the only channel-like variant for which there is a crystal structure.) With minimum radii of ~0.9 Å (**Figure 9**) these bottlenecks are still too narrow to allow Cl$^-$ permeation (r(Cl$^-$) ≈ 1.81 Å) (**Shannon, 1976**). Thus, additional opening motions in the gate region are needed for Cl$^-$ transport.

To test the idea that additional gate-opening motions occur in the absence of Glu$_{ex}$, the computational analysis discussed above was applied to characterize and analyze the collective motions of channel-like ClC-ec1. The analysis revealed that there are fewer collective motions that can open the extracellular gate after the cross-link is introduced to the channel-like mutant (P=0.001–0.070) (**Figure 10A**), whereas the intracellular gate was not significantly affected (P=0.354) (**Figure 10B**). This result is consistent with that obtained in the WT background. Taken together, our MD results suggest that the cross-link at residue 417 hinders the opening of the extracellular gate – beyond the Glu$_{ex}$ motions – in both the WT and channel-like ClC-ec1.

## Helix N connects Helix P to the extracellular gate

How are motions at Helix P transmitted to the extracellular gate? Visual inspection reveals an obvious potential transduction pathway: Helix N, which forms part of the extracellular gate (**Figure 1B**, **Figure 8A**), makes direct contacts to Helix P through side-chain packing of conserved residues in each Helix (**Figure 11A,B**). We hypothesized that disrupting these contacts would disrupt transduction of Helix-P motions to the extracellular gate, thereby abolishing the inhibitory effect of the Helix-P cross-link. To test this hypothesis, we generated Helix-N mutants F357A and L361A, in which the inter-helical coupling of motion is expected to be weakened by removing bulky side chains contributing to the contact area. The mutant transporters are slow compared to WT but retain the ability to couple Cl$^-$/H$^+$ exchange (**Figure 3—figure supplement 1**). Strikingly, the D417C cross-link only weakly inhibits L361A activity and completely fails to inhibit F357A (**Figure 11C,D**). The sluggish turnover of the F357A mutant suggests that it might be insensitive to the D417C cross-link because it is already maximally inhibited. To evaluate this possibility, we examined another slow mutant, A404L. A404 lines an intracellular "portal" for water (and hence H$^+$) entry into the transporter (**Han et al., 2014**). This residue is located at the N-terminal end of Helix P (**Figures 7A**, **11A**), which does not contact Helix N. We found that the activity of the A404L mutant, despite being similarly sluggish to F357A, is reduced further yet by the D417C cross-link (**Figure 11E,F**). Thus, the lack of sensitivity of F357A to the D417C cross-link appears due to the weakened interaction with Helix P and not to its already-low turnover. These results provide strong support for the hypothesis that Helix-P motions are transmitted to the extracellular gate via side-chain contacts to Helix N.

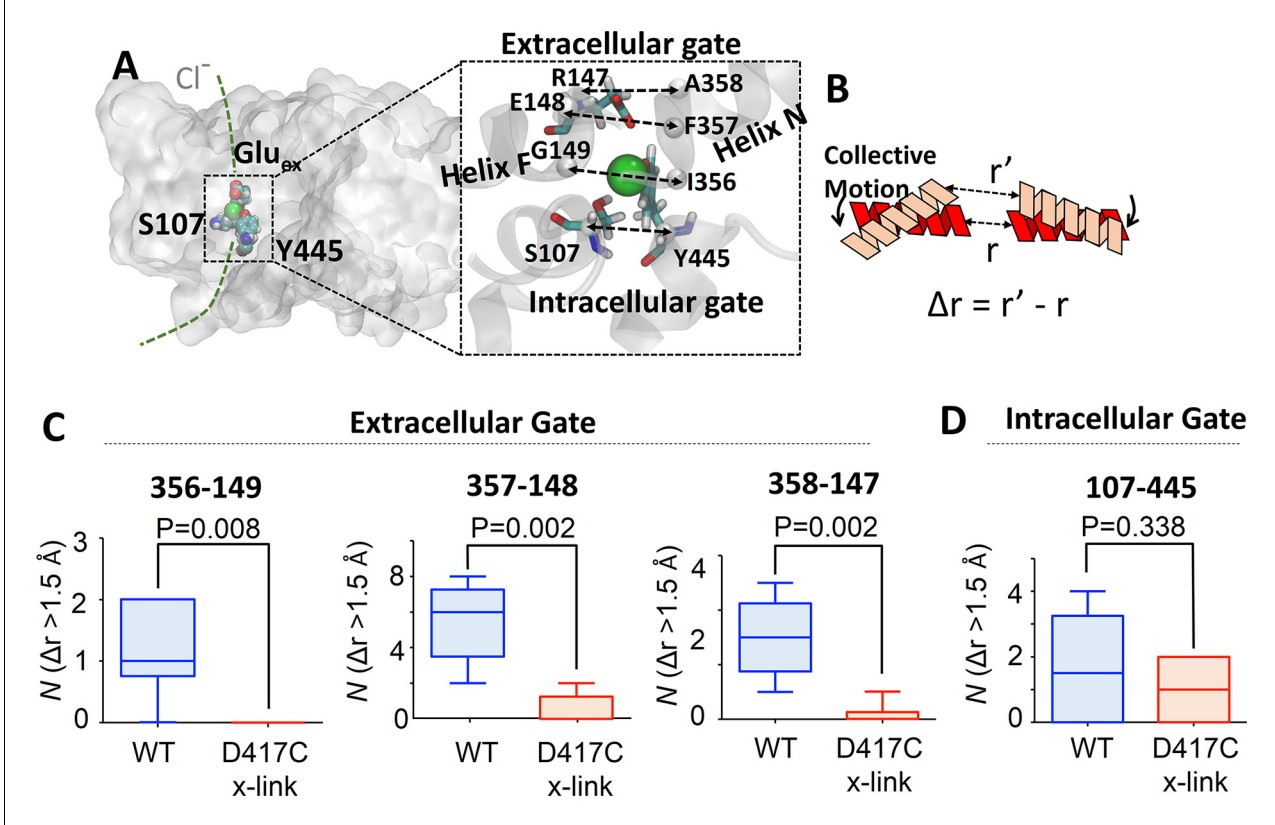

**Figure 8.** Coupling of extracellular and intracellular gating motions to collective motions in ClC-ec1 detected computationally. (**A**) Key inter-Cα distances were employed to detect functional motions. The left panel shows the location of the Cl⁻ gates (dashed box) and transport pathways (dashed green line) in ClC-ec1. Right panel shows a close-up of the Cl⁻ gates where key inter-Cα distances for both the extracellular and intracellular gates are denoted by dashed double arrows. (**B**) Scheme for determining distance change (Δr) caused by a collective motion. Following a collective motion, a native structure (red helices) undergoes structural transition (peach helices). As a result, the distance between the helices increases by $\Delta r = r' - r$. (**C**) Opening motions of the extracellular gate. The number (N) of collective motions that lead to distance changes (Δr > 1.5 Å) at each of the extracellular-gate residue pairs was determined from analysis of MD simulations for WT and cross-linked ("D417C x-link") ClC-ec1, as described in the text. The data are shown in a box-and-whisker plot where the whiskers denote minimum and maximum of the data and the box denotes the range of 25th percentile to 75th percentile of the data when sorted. The horizontal line in the box denotes the median of the data. (**D**) The number (N) of collective motions that lead to distance changes (Δr > 1.5 Å) at the intracellular gate pair 107–445 is not significantly different between WT and cross-linked ClC-ec1.

## Discussion

Our results describe a previously unidentified protein conformational state and suggest a new framework for understanding the CLC transport mechanism, introducing two key concepts. First, the structure of the E148Q mutant, with the side chain rotated away from $S_{ext}$ (*Figure 1C*) represents an "outward-facing occluded" ($OF_{occluded}$) state (*Stein and Litman, 2014*), in which bound Cl⁻ does not have full access to the extracellular solution. Second, H⁺ binding promotes an "outward-facing open" ($OF_{open}$) state, involving conformational rearrangement of Helices N and P (*Figure 11A*), that widens the extracellular ion-permeation pathway in comparison to the known crystal structures.

The first clue to conformational change at Helix P came from our NMR studies of Y419, on the short P/Q linker, where unambiguous changes in both chemical shift and solvent accessibility of ¹⁹F-labeled Y419 are observed when the pH is lowered from 7.5 to 4.5 (*Figure 3B–D*). At pH 7.5, the lack of accessibility is consistent with the crystal structure of the occluded conformational state, which depicts Y419 in a buried position. At pH 4.5, the increased accessibility of Y419 indicates a conformational state different from that captured in crystal structures. This state (with Y419 exposed to solution) is observed in channel-like ClC-ec1 at both pH 7.5 and 4.5 (*Figure 3E*). This shift in equilibrium distribution of conformational states for channel-like ClC-ec1 is useful because it enables comparison of the disulfide cross-linking of the two states, which must be done at a pH that is

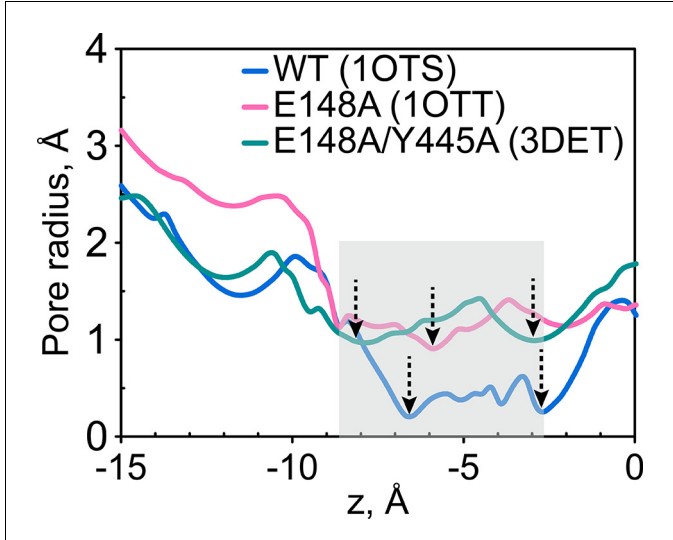

**Figure 9.** The extracellular gate remains narrow in the Glu$_{ex}$ mutant (E148A) and in the channel-like variant E148A/Y445A. The pore radius profiles of the ClC-ec1 Cl$^-$ transport tunnel for WT ClC-ec1 (blue), E148A (pink) and E148A/Y445A (green) along the z-axis (membrane normal). Shown are the profiles for subunit 1. The results for subunit 2 are very similar and thus not shown. The z-position of the central Cl$^-$-binding site is chosen as the origin of the z-axis. The shaded region denotes the extracellular-gate region; dashed arrows highlight the z-positions of the bottlenecks.

The following figure supplement is available for figure 9:

**Figure supplement 1.** Radius pore profile of 1KPL (CLC structure determined at pH 4.6).

---

amenable to disulfide bond formation (7.5 rather than 4.5). In the WT background, cross-links near Y419, at D417C, form readily (***Figure 4C***), as expected based on the crystal structure of the occluded conformational state (***Figure 5A–C***). In contrast, in the channel-like E148A/Y445S background, D417C is resistant to cross-linking (***Figure 4E***). These results suggest that the pH-dependent conformational change detected by NMR involves a change in inter-subunit proximity of D417 residues in addition to the change in solvent accessibility of Y419. DEER experiments confirm such pH-dependent change at D417 (***Figure 4G***).

Inter-subunit cross-linking of D417C restricts the conformational transition to the OF$_{open}$ state and inhibits activity. The inhibition occurs not only in the WT background but also in uncoupled E148A, Y445S, and E148A/Y445S (channel-like) backgrounds (***Figures 4*** and ***6***). Therefore, the conformational change being restricted is something other than the localized movements of side-chain gates, as these gates (E148 and Y445) are missing altogether in the uncoupled transporters. To gain insight into how conformational change near the subunit interface affects activity, we performed MD simulations on WT and channel-like ClC-ec1, with and without the D417C crosslink. We found that the major motions of both WT and channel-like involve opening of the extracellular vestibule and that these opening motions are dampened by the cross-link at D417 (***Figures 8***,***10***). Further, our mutagenesis experiments show that removing side-chain interactions between Helices N and P eliminates the effect of the cross-link on Cl$^-$ transport (***Figure 11***). Therefore, we conclude that rearrangement of these helices facilitates a widening of the extracellular ion-permeation pathway.

The residual activity remaining with maximal cross-linking at D417 (ranging from 0–300 s$^{-1}$, ***Table 1***) suggests that the OF$_{occluded}$ state may allow some minimal level of Cl$^-$ flux. However, an alternative interpretation is that the OF$_{occluded}$ is completely impermeant to Cl$^-$ and that the residual transport observed with the cross-link is either (1) not distinguishable from zero (due to compounding uncertainties in the various steps involved in the experimental measurement, including quantification of the fraction cross-linked) or (2) occurs because the cross-link does not completely prevent movement of Helix N and opening of the extracellular vestibule to the OF state. We favor the alternative interpretation as it is in keeping with the general principles of transporter function, in which

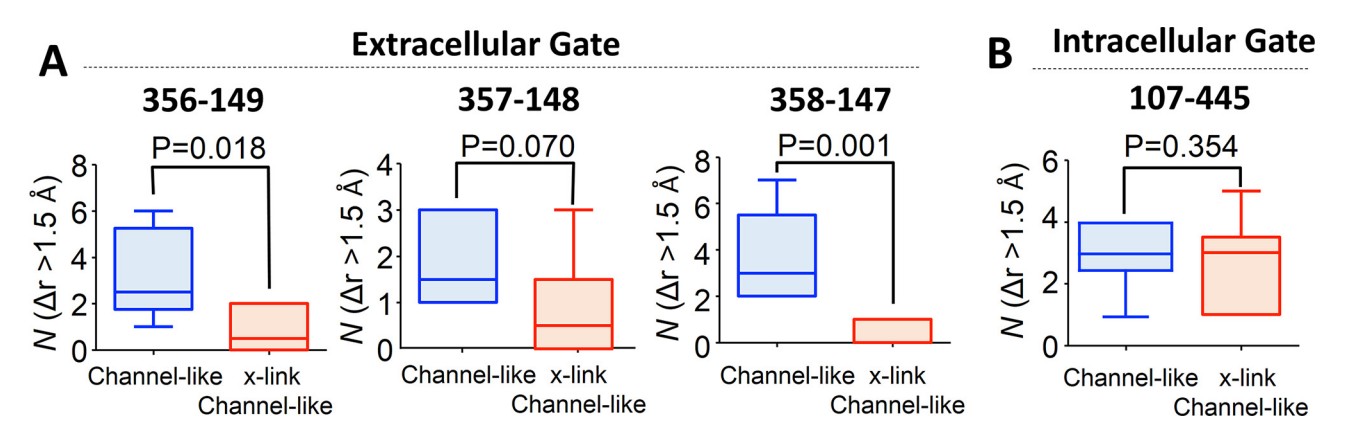

**Figure 10.** Cross-linking at 417 impedes opening of the extracellular but not the intracellular gate in channel-like ClC-ec1, as detected by computational analysis. (**A**) Opening motions of the extracellular gate. The number ($N$) of collective motions that lead to distance changes ($\Delta r > 1.5$ Å) at each of the extracellular-gate residue pairs was determined from analysis of MD simulations for WT and cross-linked ("x-link") channel-like ClC-ec1, as described in the text. The data are shown in a box-and-whisker plot where the whiskers denote minimum and maximum of the data and the box denotes the range of 25th percentile to 75th percentile of the data when sorted. The horizontal line in the box denotes the median of the data. (**B**) The number ($N$) of collective motions that lead to distance changes ($\Delta r > 1.5$ Å) at the intracellular gate pair 107–445 is not significantly different between WT and cross-linked channel-like ClC-ec1.

protein conformational change plays a key role in sustaining coupling stoichiometry. In support of this idea, we note that Helix N motions have been strongly implicated not only in ClC-ec1 (the results presented here) but also in the mammalian antiporter CLC-4 (*Osteen and Mindell, 2008*). Experiments on this homolog identified an inhibitory $Zn^{2+}$-binding site at the top of Helix N that appears to transmit conformational change to the $Cl^-$-permeation pathway at the other end of Helix N (*Osteen and Mindell, 2008*).

While it is clear that rearrangement of Helices N and P is required for opening the extracellular vestibule, the precise molecular details of this rearrangement remain to be determined. Nevertheless, several pieces of information suggest that the overall motions, though long-range in effect, may involve rearrangements/reorientations of only a few Angstroms in magnitude. First, the cross-linking of Y419C, just 5 Å away from D417C, does not inhibit function (*Figure 4—figure supplement 1*). Second, any large movement of Helix P would likely have a major effect on water entry via the narrow portal that is the rate-limiting barrier for formation of water wires and $H^+$ transport (*Lim et al., 2012*; *Han et al., 2014*). Since our computational analysis indicates that cross-linking does not significantly affect water entry (*Figure 7*), Helix-P motion may involve only a small tilt or rotation that exerts a "lever-arm" effect on Helix N and the $Cl^-$-entryway. Third, previous inter-subunit cross-linking studies targeting Helices I and Q, and the H-I and I-J loops showed that simultaneously cross-linking these regions had no significant effect on function (*Nguitragool and Miller, 2007*) and therefore argue against a major restructuring of the inter-subunit interface. Together, these results suggest that the rearrangements at Helices N and P are likely small in magnitude and do not involve the entire inter-subunit interface. This conclusion is in line with computational studies using normal-mode and functional-mode analysis, which showed the subunit interface remaining largely intact even as other regions of ClC-ec1 underwent global conformational changes to alternately expose $Cl^-$- and $H^+$-binding sites during the exchange process (*Miloshevsky et al., 2010*; *Krivobokova et al., 2012*). One of the mobile helices identified in these computational studies was Helix R, which has also been pinpointed in experimental studies of $H^+$-dependent conformational change (*Bell et al., 2006*; *Abraham et al., 2015*). Since Helix R extends from the center of the protein (where Y445 coordinates $Cl^-$, *Figure 1A*) out to the cytoplasmic solution (*Figure 1B*), the $H^+$-dependent conformational change characterized here, while not large in magnitude, may extend well beyond the immediate region around Helices P and N.

To integrate the $OF_{open}$ state into a model of the CLC transport cycle, we build on the model of Basilio et al. (*Basilio et al., 2014*). Starting with the $OF_{occluded}$ state (State 1 in *Figure 12A*, reflecting

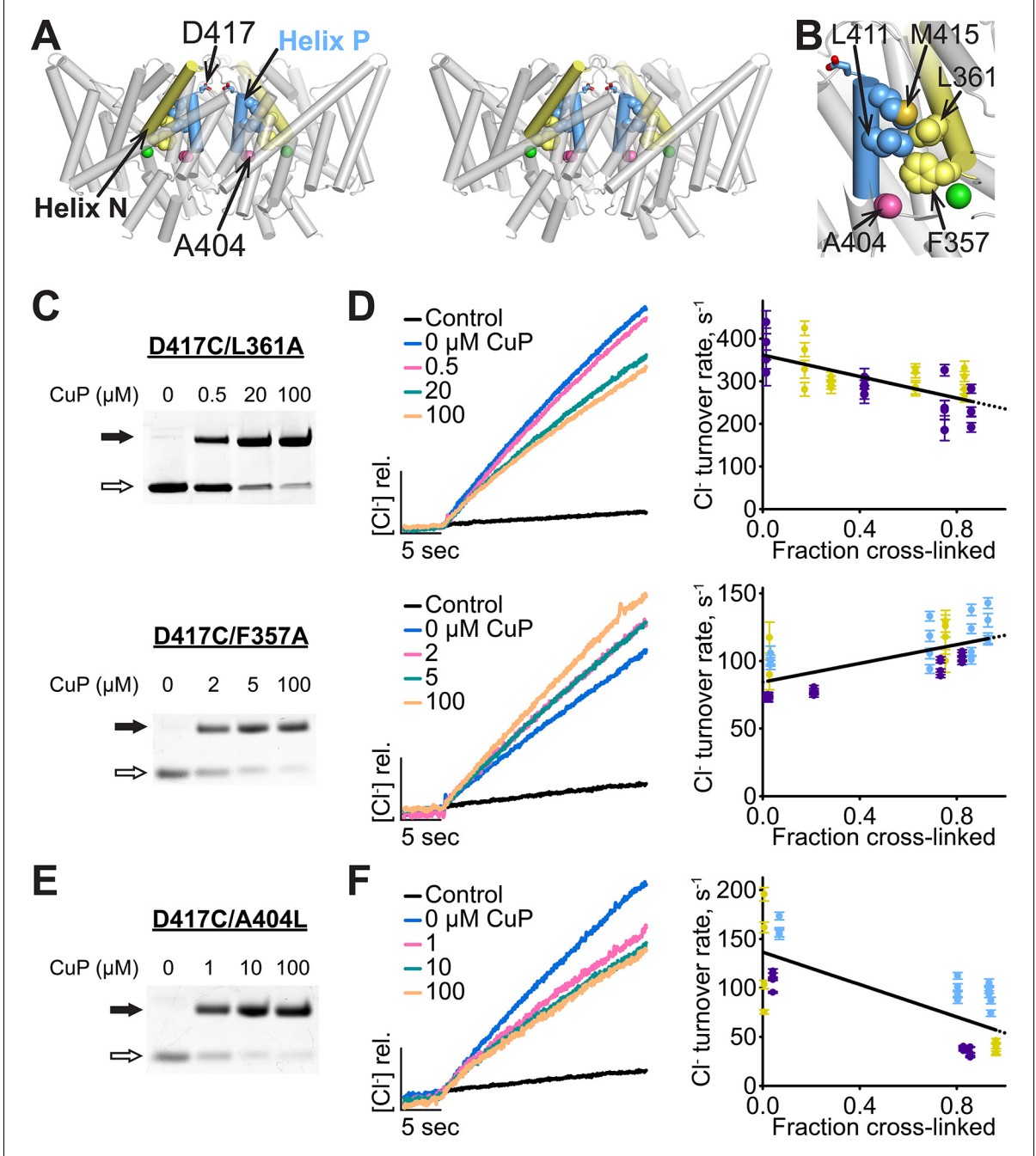

**Figure 11.** Helix P is coupled to the extracellular gate via Helix N. (**A**) Side view of ClC-ec1, in stereo. Conserved residues L411 and M415 in Helix P (blue) make direct contact with conserved residues F357 and L361 in Helix N (yellow). (**B**) Close-up of Helices P and N. (**C**) Detection of inter-subunit disulfide cross-links by non-reducing SDS-PAGE in Helix-N mutants D417C/L361A (top) and D417C/F357A (bottom). (**D**) Effect of cross-linking on activity. *Left*: Representative data traces showing Cl⁻-transport activity of D417C/L361A and D417C/F357A. *Right*: Summary data showing Cl⁻-transport activity as a function of disulfide cross-linking. Each data point represents individual data points as described in *Figure 4*. Purple, yellow and blue each represent data obtained from a separate protein preparation. (**E**) Detection of inter-subunit disulfide cross-links on D417C/A404L (**F**) Effect of cross-linking on activity of D417C/A404L. Purple, yellow and blue represent data obtained from separate protein preparations.

the state captured in the E148Q crystal structure, *Figure 1B,C*), a conformational change generates the OF$_{open}$ state (State 2). This conformational change is pH dependent (*Figures 3,4*) but need not be promoted solely by the protonation of Glu$_{ex}$, as suggested by previous observations of H⁺-

dependent conformational changes in $Glu_{ex}$ mutants (*Bell et al., 2006*; *Elvington et al., 2009*). The conformational change allows 2 $Cl^-$ ions to exit to the extracellular side (State 3). Entry of the protonated $Glu_{ex}$ into the vacated permeation pathway (State 4) facilitates transfer of one $H^+$ to the intracellular side, via water wires and the internal $H^+$-transfer site $Glu_{in}$ (*Figure 1A*) (*Accardi et al., 2005*; *Lim and Miller, 2009*; *Lim et al., 2012*; *Han et al., 2014*). Upon unbinding of $H^+$, the protein adopts the apo occluded conformation (State 5) which can then undergo conformational change to the inward-facing state (State 6, (*Basilio et al., 2014*)). Binding of 2 $Cl^-$ from the intracellular side knocks $Glu_{ex}$ out of the $S_{ext}$-binding site (State 7), which then allows $H^+$ binding from the extracellular side (back to State 1). This revised model is completely consistent with previous experimental observations, and the addition of new conformational states adds potentially key control points to the mechanism. First, the extracellular occlusion in State 7 assures no extra $Cl^-$ slips through during the step in which $Cl^-$ binds from the intracellular side. Second, we hypothesize that the $OF_{open}$ state lowers $Cl^-$ affinity and promotes $Cl^-$ release, as suggested by the increase in $Cl^-$-binding affinity observed when formation of the $OF_{open}$ state is inhibited by the D417C cross-link (*Figure 5D,E*).

Our revised model also sheds light on the mechanism of channel-like ClC-ec1. Previously, it was recognized that the narrow pathway depicted by the crystal structures of channel-like ClC-ec1 is not sufficiently wide to allow rapid ion conduction and that protein dynamics (either breathing or conformational change) must play an important part in the mechanism of ion conduction (*Jayaram et al., 2008*). Our results clarify the issue by showing that channel-like ClC-ec1 populates a conformation different from that seen in the crystal structure and exhibiting similarities to the new $OF_{open}$ state characterized in these studies. In this state, the region of the narrowest constriction – just above $S_{ext}$ – is significantly widened (*Figure 12B*). The population of this state explains why channel-like ClC-ec1 can conduct $Cl^-$ rapidly at pH 7.5.

The long-range conformational change described here improves our understanding of CLC mechanisms by providing a first glimpse of an "outward-facing open" CLC conformational state and its mechanistic implications. In future studies, it will be important to investigate the transition between the $OF_{open}$, $OF_{occluded}$ and inward-facing conformational state(s). Using a cross-linking strategy, Basilio et al. showed that transition to the inward-facing state involves motion of the intracellular half of Helix O. This motion is thought to be limited in scope, as it is relayed directly to the intracellular gate via a steric interaction between intracellular-gate residue Y445 (*Figure 2A*) and Helix O residue I402 (*Basilio et al., 2014*). Nevertheless, since Helix O also makes direct contacts to Helices N and P (studied here), it seems likely that intracellular and extracellular gate-opening motions will be linked through the interaction of these three helices. Understanding these interactions will be critical to providing a molecularly detailed view of the CLC transport mechanism.

## Materials and methods

### Expression, purification, reconstitution and flux assays

Expression and purification of unlabeled ClC-ec1 WT and mutant proteins was performed as documented in detail (*Accardi and Miller, 2004*) except that the final purification step was by size exclusion chromatography on a Superdex gel filtration column (*Walden et al., 2007*) rather than ion-exchange chromatography. Point mutations introduced by conventional PCR methods were confirmed by sequencing. D417C constructs were made in a previously characterized cysteine-less background C85A/C302A/C347S (*Nguitragool and Miller, 2007*), which here is referred to as the "WT background". For preparing ClC-ec1 under reducing conditions, 20 mM β-mercaptoethanol (β-ME) and 1 mM dithiothreitol (DTT) (Fisher Scientific, Pittsburgh, PA) were added to cell pellets during resuspension, and 1 mM DTT was included in subsequent purification steps. DTT was removed in the final purification step over a Superdex 200 size exclusion column.

To measure turnover rates in flux assays, ClC-ec1 variants were reconstituted into liposomes by dialysis (*Walden et al., 2007*) into buffer R (300 mM KCl, 40 mM Na-citrate, pH 4.5) using 0.2 – 5 μg protein per mg of *E. coli* polar lipids (Avanti Polar Lipids, Alabaster, AL). For the high-turnover channel-like variant, the lower end of this range (0.2 μg protein per mg lipids) was used. For experiments to determine stoichiometry, protein to lipid ratio was 0.4–10 μg protein per mg lipid (with higher ratios used for low-turnover mutants). Reconstituted liposomes were subjected to 4 freeze-thaw cycles and were extruded through 400-nm filters 15 times using an Avanti Mini-Extruder. Liposomes

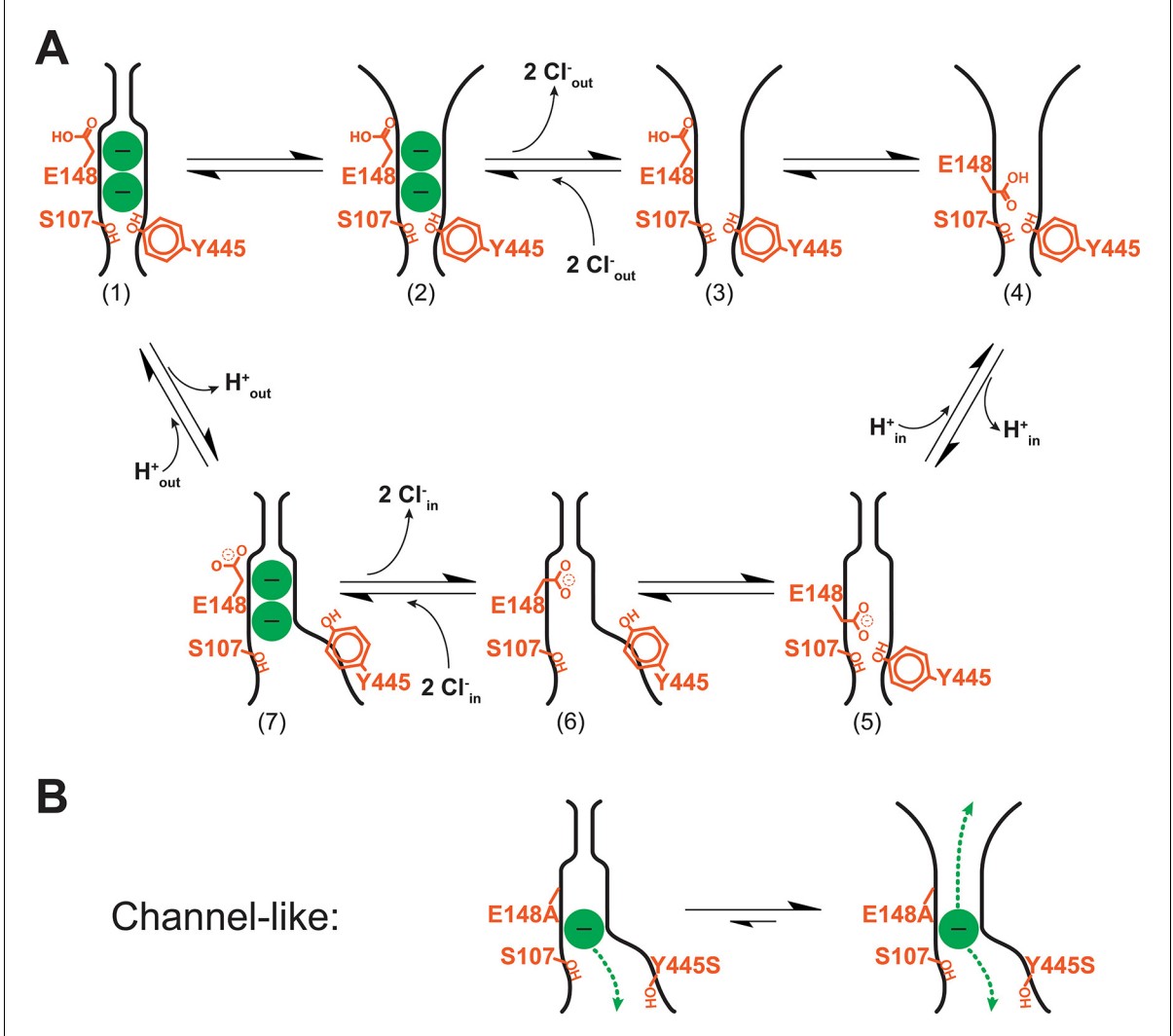

**Figure 12.** Revised model of the CLC transporter mechanism. (**A**) CLC transporter cycle. The $OF_{occluded}$ state (1) undergoes a conformational change to $OF_{open}$ (2). This step is pH-dependent but may be promoted by protonation of residues other than $Glu_{ex}$ (see Discussion). Two $Cl^-$ ions leave (3) and then entry of the protonated $Glu_{ex}$ into the permeation pathway (4) facilitates $H^+$-transfer to the inside (via $Glu_{in}$, *Figure 1B*) (5). Conformational change to the inward-facing state (6) allows 2 $Cl^-$ ions to enter from the intracellular side, knocking $Glu_{ex}$ out of the pathway (7). The cycle is reversible, with protonation favoring conformational change to the $OF_{open}$ state. (**B**) Channel-like CLC states. The crystal structure of channel-like ClC-ec1 reveals a narrow constriction at the extracellular-gate region, depicted at left. However, results here demonstrate that the major conformation adopted in solution more closely resembles the $OF_{open}$ state (equilibrium shifted to right). This finding is consistent with the high $Cl^-$ throughput observed in channel-like ClC-ec1.

were buffer-exchanged through Sephadex G-50 spin columns (*Basilio and Accardi, 2015*) into flux-assay buffer (300 mM K-isethionate, 50 µM KCl, buffered with 2 or 40 mM Na-citrate pH 4.5). (The 2 mM Na-citrate buffer was used in experiments in which $Cl^-$ and $H^+$ transport were measured in parallel; the 40 mM Na-citrate buffer was used in experiments in which only $Cl^-$ transport was measured.) Transport was initiated by addition of 2 µg/mL valinomycin (for dual $Cl^-/H^+$-transport measurements) or 3 µg/mL CCCP + 7 µg/mL valinomycin (for $Cl^-$-transport measurements) (*Han et al., 2014*). At the end of each flux-assay experiment, total liposomal $Cl^-$ was determined by disrupting the liposomes with Triton X-100 (0.01%; from a 10% stock solution); flux-assay traces shown in *Figures 4*, *6* and *11* show normalization to this value. Transport turnover rates were calculated by measuring the initial velocity of the $Cl^-$ and/or $H^+$ transport (*Walden et al., 2007*). Stoichiometry was determined from the ratio of the $Cl^-$ to the $H^+$ turnover rate. Flux assays were performed in sets of 20–40

samples; within each set, an assay was discarded if the total liposomal [Cl⁻] (a measure of the yield of reconstituted liposomes, which affects the accuracy of the unitary-turnover calculation) was >30% outside of the mean. Flux-assay measurements were performed on at least 4 samples for each condition. This sample size and selection method is based on previous experience with flux-assay measurements (*Howery et al., 2012*; *Han et al., 2014*).

## $^{19}$F NMR

$^{19}$F-Tyr labeling was performed as described (*Elvington et al., 2009*). Labeled ClC-ec1 was purified into Buffer A (150 mM NaCl, 10 mM HEPES (Fisher Scientific, Pittsburgh, PA), pH 7.5 and 5 mM n-decyl β-D maltopyranoside (DM) (Anatrace, Maumee, OH), then concentrated to approximately 50 µM. *E. coli polar lipids* were added in a 1:80 lipid:detergent molar ratio to the BuriedOnly construct to enhance stability (*Elvington et al., 2009*). The Y419Only construct was more stable without the addition of lipids. 10% $D_2O$ was added prior to NMR experiments. Samples (~300 µL starting volumes) were placed in the outer tube of Shigemi symmetrical microtubes in order to reduce the volume of sample required for data acquisition. The Shigemi tube insert was not used so as to avoid generating froth from adjusting the plunger in the detergent containing sample. Data were collected using a 5 mm H/F probe on a Bruker Avance 500 MHz spectrometer running Topspin version 1.3 with variable temperature control. Data represent acquisition of 30 – 50k transients at 470 MHz; 12 kHz spectral width; 45° pulse; 0.17s acquisition time; 1.8 – 2.8s relay cycle; 20°C; 15 Hz linebroadening; referenced to TFA. The pH of the samples was lowered to 4.5 using a 1 M citric acid solution (EMD Millipore, Billerica, MA) and raised to 7.5 using a 1M Tris-acetate pH 9.0 solution. TEMPOL (4-Hydroxy-2,2,6,6-tetramethylpiperidine 1-oxyl, Fluka Analytical, Ronkonkoma, NY) was added to the sample by carefully weighing out and adding the solid reagent required to attain a final concentration of 100 mM in the NMR sample.

## Cysteine cross-linking

All procedures were carried out at room temperature (21–23°C). Stock solutions of CuP at 10x were made from 1:3 mixtures of $CuSO_4$ (aqueous) (MCB Reagents, Cincinnati, OH) and 1,10-phenanthroline (in ethanol) (Sigma-Aldrich, St. Louis, MO). ClC-ec1 eluted from the Superdex 200 column in Buffer A was diluted to 0.2 mg/mL (1.9 µM homodimer; 3.8 µM Cys-containing subunits) before addition of CuP. After an hour of incubation, 1 mM Na-EDTA (Fisher Scientific, Pittsburgh, PA) was added to terminate the cross-linking reaction. Cross-linking was visualized using SDS/PAGE (4–15% gradient gels) and staining with Coomassie brilliant blue (TCI America, Portland, OR). Cross-linking was documented using an Odyssey Infrared Imaging System (LI-COR Biosciences) using the 700 nm channel. ClC-ec1 band intensities were quantified using NIH ImageJ software. Un-cross-linked ClC-ec1 runs as a monomer (apparent molecular weight ~36 kD) and cross-linked mutant as a dimer (apparent molecular weight ~64 kD). The fraction cross-linked was calculated based on the relative intensities of the dimer and monomer bands. For channel-like ClC-ec1, which exhibited a low efficiency of cross-linking, free thiols were quantified colorimetrically (Life Technologies Thiol and Sulfide quantification kit, T6060). During reconstitution into liposomes, most CuP-treated samples were dialyzed in buffer containing 1 mM DTT in order to avoid additional cross-linking during the dialysis step; this level of DTT was sufficiently low that it did not reduce D417C disulfide bonds that had already been formed. Mutant proteins D417C/F357A and D417C/L361A were reconstituted in the absence of DTT when crosslinked with 100 µM CuP, as an extra precaution to avoid disulfide-bond reduction in these samples.

## DEER

For preparing D417C ClC-ec1 (WT and channel-like backgrounds) for DEER experiments, 20 mM β-ME was added to cell pellets during resuspension. β-ME was removed during washing and elution from the cobalt column. Proteins eluted from the cobalt column were incubated with 50x molar excess of the paramagnetic spin label MTSSL (Enzo Life Sciences, Farmingdale, NY) that was dissolved in small volume of dimenthylformamide (DMF) (Fisher Scientific, Pittsburgh, PA) such that the final DMF concentration was < 0.1%. The protein solution was sealed under argon and mixed by slow rotation (~20 rpm) for 2 hr at room temperature. The remaining steps of the purification were identical to our usual ClC-ec1 purifications. Thus, the 6-His tag was then removed by a one-hour

incubation with endoprotease Lys-C (Roche Diagnostics, Indianapolis, IN). The ClC-ec1 samples were then purified from the cleaved 6-His tag and excess MTSSL by size exclusion chromatography on a Superdex 200 column. Glycerol (23% v/v) was added to the protein solution as cryoprotectant. This was achieved by adding an 80% (v/v) glycerol stock solution (prepared in buffer A) to the purified protein. The samples were then concentrated to a final concentration of 50–100 μM, and *E. coli* polar lipids were added at 1:80 lipid:detergent molar ratio. A stock solution of 25 mM citrate was used to adjust the sample at pH 7.5 to pH 4.5. Functional assays were performed on EPR samples that had been exposed to pH 4.5 for one hour before reconstitution. CW-spectra were collected on a Bruker EMX at 10 mW power with a modulation amplitude of 1.6G. Spectra were normalized to the double integral. DEER experiments were carried out using a standard four-pulse protocol (*Jeschke, 2012*). Samples were maintained at 83K. DEER distributions were obtained from fitting the DEER decays to a sum of Gaussian distributions (*Brandon et al., 2012*; *Mishra et al., 2014*; *Stein et al., 2015*).

## Structure determination

For crystallization, the D417C mutant was put into a deletion construct (ΔNC) lacking N-terminal residues 2–16 and C-terminal residues 461–464 (*Lim et al., 2012*). Purified ΔNC-D417C was cross-linked with 100 μM CuP for 1 h, incubated with excess Fab fragment (*Dutzler et al., 2003*) for 30 min, then purified by size exclusion chromatography (Superdex 200) into buffer containing 100 mM NaCl, 5 mM DM, 10 mM Tris (Fisher Scientific, Pittsburgh, PA), pH 7.5. The complex was concentrated to 10–12 mg/mL and mixed with 30% PEG 400 (Hampton Research, Aliso Viejo, CA), 0.075 M K/Na-tartrate (Fluka Analytical, Ronkonkoma, NY), 0.1 M Tris HCl (MP Biomedicals, Santa Ana, CA) (pH 9.0). Crystals were grown by the sitting drop method for 2–4 weeks at 20°C and were directly harvested from the reservoir, flash frozen and stored in liquid $N_2$. Diffraction data were collected to 0.9795 Å at the BL12-2 beamline (SLAC) and processed using XDS (*Kabsch, 2010*). Phases were obtained by molecular replacement with the WT protein in complex with Fab (PDB 1OTS) using the MOLREP program (*Vagin and Teplyakov, 2010*). Refinement was done using the refmac program (*Murshudov et al., 1997*). Atomic coordinate and structure factors are deposited in the Protein Data Bank under accession code 5HD8.

## Isothermal titration calorimetry

ITC was carried out using a MicroCal VP-ITC instrument. Chloride binding to WT and mutants were carried out as described previously (*Picollo et al., 2009*; *Howery et al., 2012*). Briefly, ClC-ec1 (WT or D417C or D417 cross-linked using 100 μM CuP) was purified over a Superdex 200 size exclusion column pre-equilibrated with Buffer B (100 mM $K^+$-$Na^+$-tartrate, 20 mM HEPES, 5 mM DM, pH 7.5) and then concentrated to 25–50 μM. Percent cross-link following treatment with 100 μM CuP was 92.0 ± 0.6% (n=2). The injection syringe was filled with Buffer B containing 20 mM KCl. Each experiment consisted of 30 10-μL injections of the $Cl^-$-containing solution at 5 min intervals, to achieve a final molar ratio of 50–160. The chamber was kept at 25°C with constant stirring at 350 rpm. All solutions were filtered and degassed before use. ITC data were fit to a single-site isotherm as described with Origin 7 MicroCal program.

## Molecular dynamics (MD) simulations

The ClC-ec1 crystal structure at 2.51 Å (PDB ID: 1OTS) (*Dutzler et al., 2003*) was used to prepare for the MD simulations of all the systems studied in the present work – WT, D417C, channel-like (E148A/Y445S), and D417C/channel-like. The system setup for the WT ClC-ec1 is detailed in our previous work (*Han et al., 2014*). In short, to have the protein hydrated properly, all the crystallographic water molecules were maintained and 49 additional water molecules were added using DOWSER (*Zhang and Hermans, 1996*). One additional water molecule was placed between $Glu_{ex}$ (E148) and the $Cl^-$ ion bound to the central ion-binding site of ClC-ec1 (*Figure 1*) in order to stabilize the two closely (within ~4 Å) positioned negative charges, as suggested in previous simulation studies (*Bostick and Berkowitz, 2004*; *Cohen and Schulten, 2004*; *Wang and Voth, 2009*). $Glu_{ex}$ (E148) and $Glu_{in}$ (E203) were both deprotonated, while E113 was modeled in its protonated form according to previous Poison-Boltzmann electrostatic calculations (*Faraldo-Gomez and Roux, 2004*). The

protein was embedded into a POPE lipid bilayer, fully equilibrated TIP3P water (*Jorgensen et al., 1983*) and buffered in 150 mM NaCl, resulting in a $105 \times 105 \times 110$ Å$^3$ box with ~110,000 atoms.

The mutant systems were constructed on the basis of that of the WT. For each mutant, residue substitutions were done using the Mutator plugin of VMD (*Humphrey et al., 1996*). Disulfide bonds were constructed by introducing geometric restraints on two cysteine residues, including a distance restraint between the sulfur atoms and angular restraints involving C$_\beta$ atom of either cysteine and the two sulfur atoms. To avoid structural disruption of the protein due to sudden introduction of restraints, the disulfide restraints were turned on gradually over 20-ns simulations. Note that the systems prepared as such are not significantly different from the cross-linked D417C crystal structure. In fact, during the equilibrium simulation of the mutant containing the disulfide bond (see below), the RMSD of the protein to the D417C crystal structure is on average ~1.6 Å, even smaller than its RMSD (~1.9 Å) to the WT crystal structure that the simulation started from.

All MD simulations were carried out with NAMD 2.9 (*Phillips et al., 2005*) using the CHARMM-CMAP (*Mackerell et al., 2004*) and CHARMM36 force fields (*Klauda et al., 2010*) to model the proteins and lipids, respectively. The particle mesh Ewald (PME) (*Darden et al., 1993*) method was used to calculate long-range electrostatic forces without truncation. All simulation systems were subjected to Langevin dynamics and the Nosé-Hoover Langevin piston barostat (*Nose, 1984*; *Hoover, 1985*) for constant pressure (P = 1 atm) and temperature (T = 310 K) (NPT). Each system was energy-minimized for 5,000 steps, followed by a 1-ns MD run with positions of all protein atoms and oxygen atoms of the crystallographic water molecules restrained. Each system was simulated without any restraints for ~300 ns.

## Analysis of collective motions of protein

The collective motions of the protein were analyzed through principal component analysis (PCA) of the equilibrium MD trajectories (*Amadei et al., 1993*). Specifically, we first constructed the covariance matrix $C$ of C$_\alpha$ atoms of select parts of the proteins for each subunit based on equilibrium MD trajectories. The covariance matrix $C$ was calculated as $c_{ij} = <(x_{in} - <x_{in}>)(x_{jn} - <x_{jn}>)>$, where $X_n=\{x_{in}\}$ are the coordinates of C$_\alpha$ atoms of select parts of protein in the n$^{th}$ sampled structure and the brackets <> denote the averages over all the sampled structures. The first 50 ns of each MD trajectory were discarded to remove any initial bias. Only the transmembrane helical regions were selected for this analysis as they define the overall architecture of the protein and most relevant to the functionally relevant global motions. We then derived orthonormal eigenvectors $R=\{R_k\}$ of the covariance matrix $C$. Each eigenvector $R_k=\{r_{ik}\}$ defines relative movement ($r_{ik}$) of each select atom in a collective motion of the protein represented by the eigenvector. The 20 eigenvectors with the largest eigenvalues were chosen for further analysis. These eigenvectors correspond to the collective motions that account for >75% of protein motion observed in the simulations.

Following the approach by Bahar and co-workers (*Isin et al., 2008*), conformational deformation driven by a given collective motion can be calculated according to the associated eigenvector $R_k$ as follows:

$$X' = X_0 \pm AR_k \qquad (1)$$

where $X_0$ and $X'$ denote the coordinates of the reference structure and the structure of the protein deformed by the collective motion, and $A$ is an arbitrary scaling factor determining the extent of structural deformation to be examined. The value of $A$ is related to the RMSD between the reference and the deformed structures through the relationship RMSD = $A/M^{1/2}$, where $M$ is the number of atoms selected to calculate RMSD (here $M$=538, the number of C$_\alpha$ atoms located in the transmembrane helical region of the protein). To make a meaningful comparison of all collective motions investigated, the value of $A$ was chosen such that the structure of the protein is altered by each motion to the same extent, targeting always a total RMSD of 3.5 Å with respect to the original structure. Thus, the distance change ($\Delta r$) between two sites ($x_i$ and $x_j$) of interest (*Figure 8B*) can be calculated according to $\Delta r = |x'_i - x'_j| - |x_i - x_j|$. Finally, we quantified the protein's ability of opening its gates via collective motions by counting the dominant collective motions that involved an increase in the distance between residues lining the gates by $\Delta r > 1.5$ Å. To achieve a statistical estimate of such counts, each ~300-ns simulation trajectory of the homodimer was divided evenly into three time blocks (*Rapaport, 2004*), each being analyzed through the procedure described above,

providing a dataset of six segments (three time blocks for each subunit x 2 subunits). Statistical comparisons between datasets were made using the Wilcoxon-Mann-Whitney test (*Mann and Whitney, 1947*).

## Acknowledgements

We thank Martin Prieto for comments on the manuscript. We thank Hyun-Ho Lim, Carole Williams and Chris Miller for the hybridoma producing the antibody used for crystallization and for the cDNA encoding ΔNC ClC-ec1. We thank Rudi Nunlist and Dr. Christian Canlas, College of Chemistry NMR facility, University of California at Berkeley, for use of the H/F NMR probe. We thank Chris Garcia and Michael Birnbaum for use of the MicroCal ITC instrument. All simulations have been performed using XSEDE resources (grant number MCA06N060). Use of the Stanford Synchrotron Radiation Lightsource, SLAC National Accelerator Laboratory, is supported by the U.S. Department of Energy, Office of Science, Office of Basic Energy Sciences under Contract No. DE-AC02-76SF00515. The SSRL Structural Molecular Biology Program is supported by the DOE Office of Biological and Environmental Research, and by the National Institutes of Health, National Institute of General Medical Sciences. We are grateful to Stanford's Chemical Biology Institute for birdseed funding to produce antibody used in this study.

## Additional information

### Funding

| Funder | Grant reference number | Author |
| --- | --- | --- |
| National Science Foundation | 1021472 | Merritt Maduke |
| National Institutes of Health | U54-GM087519 | Hassane S Mchaourab<br>Emad Tajkhorshid<br>Merritt Maduke |
| Stanford University | Chemical Biology Institute Birdseed Funding | Merritt Maduke |
| G Harold and Leila Y. Mathers Foundation | Research Award | Merritt Maduke |
| National Institutes of Health | GM086749 | Emad Tajkhorshid |
| National Institutes of Health | P41-GM104601 | Emad Tajkhorshid |
| XSEDE | | Emad Tajkhorshid |
| U.S. Department of Energy | DE-AC02-76SF00515 | Irimpan I Mathews |

The funders had no role in study design, data collection and interpretation, or the decision to submit the work for publication.

### Author contributions

CMK, Performed cross-linking experiments, Produced protein samples, Reconstituted into liposomes, Performed functional assays, Performed ITC experiments, Acquisition of data, Analysis and interpretation of data, Drafting or revising the article; SJA, Performed cross-linking experiments, Produced protein samples, Reconstituted into liposomes, Performed functional assays, Performed NMR experiments, Performed ITC experiments, Acquisition of data, Analysis and interpretation of data, Drafting or revising the article; WH, Performed computational simulations and analyses, Conception and design, Acquisition of data, Analysis and interpretation of data, Drafting or revising the article; TJ, Performed computational simulations and analyses, Acquisition of data, Analysis and interpretation of data, Drafting or revising the article; TSC, RCC, Performed cross-linking experiments, Produced protein samples, Reconstituted into liposomes, Performed functional assays, Acquisition of data, Analysis and interpretation of data, Drafting or revising the article; SME, CWL, Performed NMR experiments, Acquisition of data, Analysis and interpretation of data, Drafting or revising the article; IIM, Determined the structure of D417C ClC-ec1, Acquisition of data, Analysis and interpretation of data, Drafting or revising the article; RAS, Acquired and analyzed the DEER/

EPR data, Acquisition of data, Analysis and interpretation of data, Drafting or revising the article; HSM, Supervised the DEER/EPR experiments, Analysis and interpretation of data, Drafting or revising the article; ET, MM, Designed experiments and supervised the work, Conception and design, Analysis and interpretation of data, Drafting or revising the article

### Author ORCIDs

Emad Tajkhorshid, http://orcid.org/0000-0001-8434-1010

## Additional files

### Major datasets

The following datasets were generated:

| Author(s) | Year | Dataset title | Dataset URL | Database, license, and accessibility information |
|---|---|---|---|---|
| Irimpan II, Khantwal CM, Maduke M | 2016 | Crystal structure of disulfide cross-linked D417C ClC-ec1 | www.rcsb.org/pdb/explore/explore.do?structureId=5HD8 | Publicly available at the Protein Data Bank (accession no. 5HD8). |

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
