## [Decision Letter]

Thank you for submitting your work entitled "Conformational change in CLC transporters: beyond the rotation of Glu_ex_" for peer review at *eLife*. Your submission has been favorably evaluated by Richard Aldrich (Senior editor and Reviewing editor) and three reviewers.

The reviewers have discussed the reviews with one another and the Reviewing editor has drafted this decision to help you prepare a revised submission.

Summary:

This manuscript describes a multimethodological attack on a longstanding dirty little secret hanging over the field of structure-mechanism analysis of CLC transporters and channels. From the first structures over a decade ago, the notion has emerged that a key event in the coupling of anion and proton movements is the sidechain rotation of a conserved "external Glu" residue (Glu_ex_) that opens a pathway for Cl^-^ to the extracellular solution, thus converting an "ion-occluded" conformation to an outside-open one. The mechanistic centrality of this Glu_ex_ movement is not in doubt, but the precise nature of the outward-open pathway has been mostly ignored, probably because anyone who has actually examined CLC structures can immediately see that it's got a problem: the pathway in structures of the presumed outward-open conformation connecting the central bound Cl^-^ ion to external solvent is far too narrow for the ion to squeeze through. The necessity of "protein breathing" has been occasionally invoked, in a more-or-less handwaving spirit, to dismiss this issue as the field builds various models for anion-proton antiport.

The work takes on the problem directly by bringing a combination of ^[19]^F NMR, crystallography, crosslinking, and molecular dynamics to bear on the question: does outward opening involve conformational movements more extensive than a mere Glu_ex_ rotameric switch? Recent indicators that some sort of backbone rearrangement may be involved have previously been reported from Maduke's lab, using ^[19]^F NMR, and from Accardi's using crosslinking. Now the authors bear down on the problem in more detail. They identify a residue, buried in the known structures and distant from Glu_ex_, that appears to become water-exposed at low pH, where pathway opening occurs. Inter-subunit disulfide crosslinking of this position does not affect transport, but this 'negative' result is proposed to reflect flexibility of the loop on which the residue resides. They then test a nearby residue on TM Helix P, and find that crosslinking can also occur here, and that the extent of crosslinking (under fairly harsh oxidation conditions) is tightly correlated with transport inhibition. The proposal naturally emerges that pathway opening involves a movement of TM Helices in this region. The idea is then examined by molecular dynamics, which look for collective motion tendencies that would widen the external pathway. Such motions are found in TM Helices F and N, the latter of which contacts Helix P. These motions are suppressed in MD simulations of the crosslinked structure (directly determined by a crystal structure). Mutations that decrease steric contact between Helices N and P are then tested and are found to confirm the prediction that such a loss of contact should remove the inhibitory effect of crosslinking on transport.

This set of independent lines of analysis, any one of which would alone be unconvincing, lead to a picture that should now be taken seriously by the field: that rather subtle movements of helices distant from Glu_ex_ lead to a low-pH conformation slightly different from those repeatedly observed in crystal structures (nearly all of which are at pH 7-9, but see below). In this conformation, as visualized by the MD simulations, the external pathway would open an additional 2 A, which would allow bound Cl^-^ access to the extracellular solution. These suggested motions are small – much smaller than the 'rocking-banana' or 'elevator' conformational changes involved in conventional alternating-access mechanisms of other transporters – and the general view of a nearly 'solid-state' mechanism of antiport in CLCs remains valid in broad outline.

The problem addressed in the manuscript is important, the results interesting, the design of high quality and the manuscript is well written. However, enthusiasm is dampened as there is no single piece of evidence that is completely convincing on its own. While the various parts are of high quality, they are also circumstantial and require extensive hand-waving arguments to hold together. There is no "nail in the coffin" type experiment that unambiguously proves their point. Furthermore, key experiments are missing and some of the results are inconsistent with the authors' interpretation and model. These inconsistencies weaken the logical connections between the observations and the authors' interpretation of the data. The following issues need to be addressed to clarify and strengthen the arguments.

Essential revisions:

1) The recorded NMR signals appear to an untrained eye weak and noisy. While we understand that this is unavoidable when working with detergent-solubilized large membrane proteins, we wonder how do the authors determine whether the differences between spectra are significant or are within the error of the experiment? It would be great if the authors commented on the decision-making and on how reproducible are the observed differences.

2) In the analysis of MD results in Figure 8 and Figure 10, how does one evaluate whether differences between cross-linked and uncross-linked subunits are significant or not? For example, in Figure 10 right panel, the extracellular gates in two subunits show behaviors that by eye looks just as distinct as those of the cross-linked and uncross-linked subunits in Figure 8. Could such variations be simply due to a chance? In other words, what statistical analysis, if any, can be applied to give numbers to the terms that the authors use: "dampened", "not significantly dampened" (in description of Figure 8) and "dampened to a lesser extent" (in description of Figure 10)?

3) To justify the observation that the channel-like variant of CLC-ec1 is inhibited by the D417C crosslink, the authors argue that a relaxation of the Cl^-^ pathway, besides the movement of E148, is necessary for Cl^-^ movement. However, the evidence supporting this notion is minimal, just small movements seen in the MD simulations, and it is not clear whether these movements are indeed relevant for transport or not. It is therefore important to show that the NMR spectrum of the crosslinked channel-like mutant is different from that of the uncrosslinked one and more similar to that of the WT protein. If this were not possible (because the fraction of crosslinked channel-like mutant is too small) then the authors could test whether the spectrum of the crosslinked WT protein loses pH sensitivity. This experiment, in addition to supporting the authors' hypothesis of a conformational change in the pore, would also provide a much needed link between the conformational changes observed via the NMR to those prevented by the crosslink.

4) The authors' finding that the crosslink specifically inhibits the Cl^-^ pathway while leaving the H^+^ pathway largely unaltered is extremely troubling and in stark contrast to their transport model. If H^+^ and Cl^-^ transport are tightly coupled then locking the transporter in one conformational state should result in parallel inhibition of both fluxes. The authors show that this is not the case for the Y445S mutant. The consequence of this observation is that the two processes are not tightly coupled and that H^+^ transport can occur independent of Cl^-^ movement and of any conformational changes occurring in the protein. This finding poses critical issues that need to be resolved:

i) As far as we understand, the H^+^ fluxes are uphill transport not efflux. Where does the energy driving H^+^ movement come from if there is no Cl^-^ movement? The only ion out of equilibrium is Cl^-^, all others are at equilibrium. So if there is H^+^ movement it must be coupled to Cl^-^. It could be argued that the crosslink "rescues" the mutation by acting on the intracellular gate, reducing leak or even making the transporter "super-coupled". However this is incompatible with other parts of their arguments. Furthermore, this does not solve the issue of what is the energy source for H^+^ transport extrapolated to the "complete" crosslink conditions and inhibition. Thus, this result is puzzling from a thermodynamic point of view.

ii) The stoichiometry of the Y445S mutant is ~39:1, yet the authors see that this mutant mediates ~40% of WT's H^+^ transport rather than ~5% which would be expected for such an altered stoichiometry (assuming that the rate of the mutant is not much different from WT's). This should be clarified.

iii) In all proposed models (including the authors') H^+^ transport requires a movement of E148 which, in turn, is coupled to movement of Cl^-^ in the pore. How can this be reconciled with the finding that H^+^ transport can occur with no Cl^-^ movement and with no conformational changes?

iv) Protonation of E148 is energetically linked to Cl^-^ binding. How can E148 become protonated/deprotonated if no Cl^-^ is being transported?

5) The authors use CuP to promote crosslinking. However, they find that a large fraction of the thiols becomes oxidized and thus prevented from forming disulfide bonds, which then prevents formation of the crosslink. Have the authors tried other strategies to induce disulfide formation? Maybe ones that do not rely on the generation of local concentrations of reactive oxygen like CuP does? Since the inhibition of the crosslinked D417C mutant is the key piece of evidence linking the observed conformational changes to transport it is important to directly show that a completely crosslinked transporter is indeed fully inhibited.

6) The authors state that "The question remains whether and how these conformational changes are involved in regulating ion binding and translocation during Cl^-^/H^+^ transport." This question is not as open as the authors suggest. Basilio et al. showed that such conformational changes do occur and that they are strictly required for transport.

7) The authors show (subsection “H^+^-dependent accessibility of Y419 “and Figure 3) that the NMR peak corresponding to Y419 shifts and splits when pH is lowered. The authors interpret this (together with other evidence) as an indication that CLC-ec1 adopts different conformations. Can they use the relative weight of the peaks to estimate what fraction of the time CLC-ec1 spends in the OF state? The concern here is that low pH promotes turnover of the transporter and therefore it is plausible that rather than stabilizing a specific state this maneuver populates multiple states.

It is hard to follow the authors' reasoning when they conclude (subsection “H^+^-independent accessibility of Y419 in channel-likeClC-ec1”) that "In channel-like E148A/Y445S, the accessibility of Y419 to TEMPOL indicates that the external Cl^-^-permeation pathway of the channel-like ClC-ec1 variant adopts a conformation in solution different from that observed in the crystal structure and similar to the conformation adopted by WT at low pH." Y419 is located quite far from the ion transport pathway, so while their data indicates that the environment surrounding this residue is indeed different in the WT and channel-like backgrounds, we do not see how that the same argument can be readily extended to the Cl^-^ pathway.

8) A point omitted here which seems might go to the heart of the issue is that the PDB includes a single CLC crystal structure at pH 4.6, where the alternative outward-occluded structure of this manuscript is proposed to exist – the original 3.0 A structure of CLC from *Salmonella* (PDB 1KPL). We wonder if you see any differences in helices P, N, or F in this 'Glu_ex_-down' that might differ from the high-pH *E. coli* CLC structures? Any hints here?

9) We think that the authors are underinterpreting the crucial Figure 4 data – the correlation of fraction of protein crosslinked and fractional inhibition of transport. They ask in the Discussion (fourth paragraph): why doesn't crosslinking completely inhibit activity? – and then they go into some handwaving to suggest various possibilities. But in Figure 4 it could be concluded that to a first approximation, crosslinking fully inhibits. Granted, you cannot achieve 100% crosslinking because of the harsh CuP conditions, which, as you show, produce higher oxidation states of the thiol; but the linear correlation drawn on the figure suggests that if you could have gotten to 100%, you'd have less than 20% activity left, which might not be distinguishable, given the assays, from zero. It's of course always virtuous to be conservative in data interpretation, but this possibility ought to be mentioned along with the rather anodyne alternatives offered up to explain the 'residual activity' of the crosslinked protein.

10) More on Figure 4. You rather do yourselves a disservice by presenting the flux data as you do. Your point is to illustrate the effect of crosslinking on the initial rate of transport – i.e. at early times. But the bulk of the figure is taken up, purposelessly, by the big jump by detergent that gives you the normalization value. we would leave out that big jump, expand the y-axis to put the early-time flux traces on a full-scale display, and just tell us in the legend or Methods, that fluxes are normalize to values obtained after dissolving the liposomes with Triton. This is not merely a matter of display-aesthetics; the reader would benefit from actually being able to see the degree of inhibition, which is hard to perceive with the meaningful parts of the traces all compressed into the lower 20% of the figure.

11) Figure 4. Why do the efflux traces level off at only half the liposomes at the protein density you are using? I would expect all the liposomes to contain active CLCs.

12) In the MD simulations it would be good to know a bit more about how the special motion-types were identified. Specifically, the word "extrapolated" (subsection “Potential gate-opening motions inWT and cross-linkedClC-ec1”) in finding the looked-for collective motions raises skepticism about what, exactly, those Δ-r distances are in Figure 8. Do they represent actual frames in which the pathway is seen to be 2 A wider, or some sort of widening imagined by some sort of sophisticated extrapolation algorithm?

13) We would take issue with the cavalier use of the term "large-scale motions" to describe the sorts of wiggling you are envisioning. Of course, use of this phrase is subjective, but we think that most structural biologists would refer to these as 'breathing motions.' To call them large-scale might be viewed by many as tendentious.

---

## [Author Response]

*Essential revisions: 1) The recorded NMR signals appear to an untrained eye weak and noisy. While we understand that this is unavoidable when working with detergent-solubilized large membrane proteins, we wonder how do the authors determine whether the differences between spectra are significant or are within the error of the experiment? It would be great if the authors commented on the decision-making and on how reproducible are the observed differences.*

Yes, we agree that the challenge of working with detergent-solubilized large membrane proteins results in NMR spectra that appear weak and noisy. In our previous work (Elvington et al. EMBO J, 2009), the ^[19]^F NMR signal was stronger because we studied samples with multiple ^[19]^F-labels per ClC-ec1 subunit. The samples in this study only have a single ^[19]^F label per subunit so the weaker signal is expected. Despite the low signal, we are confident in the spectral differences observed in this work because (1) we repeated the experiments and found the results to be reproducible (see new figure: Figure 3—figure supplement 2) and (2) our observation of a pH-dependent chemical shift change in the Y419 only mutant, but not in the Y419only/channel-like mutant, is consistent with the ^[19]^F NMR results published in the previous work that showed a pH-dependent chemical shift change at Y419 in WT but not the channel-like ClC-ec1 in the multi-labeled ‘Buried Only’ ClC-ec1 background (which has better signal:noise).

*2) In the analysis of MD results in Figure 8 and Figure 10, how does one evaluate whether differences between cross-linked and uncross-linked subunits are significant or not? For example, in Figure 10 right panel, the extracellular gates in two subunits show behaviors that by eye looks just as distinct as those of the cross-linked and uncross-linked subunits in Figure 8. Could such variations be simply due to a chance? In other words, what statistical analysis, if any, can be applied to give numbers to the terms that the authors use: "dampened", "not significantly dampened" (in description of Figure 8) and "dampened to a lesser extent" (in description of Figure 10)?*

We thank the reviewers for this suggestion. To more quantitatively compare the MD results on cross-linked and uncross-linked ClC-ec1, we counted dominant collective motions that can lead to an increase in the distance between residues lining the gates by Δr > 1.5 Å. To statistically evaluate these counts, each individual extended simulation trajectory of the ClC-ec1 homodimer was divided evenly into three blocks (ref: chapter 4, p. 86, DC Rapaport, The Art of Molecular Dynamics Simulations (second edition) 2004, Cambridge University Press), providing a dataset of six segments (three for each subunit x 2 subunits). Statistical comparisons between datasets were made using the Wilcoxon-Mann-Whitney test (Annals of Mathematical Statistics 1947, 18, 50-60). The revised manuscript contains a description of this analysis (pages 30-31) and revised versions of Figure 8 and Figure 10, to show the statistical analysis.

*3) To justify the observation that the channel-like variant of CLC-ec1 is inhibited by the D417C crosslink, the authors argue that a relaxation of the Cl^-^ pathway, besides the movement of E148, is necessary for Cl^-^ movement. However, the evidence supporting this notion is minimal, just small movements seen in the MD simulations, and it is not clear whether these movements are indeed relevant for transport or not. It is therefore important to show that the NMR spectrum of the crosslinked channel-like mutant is different from that of the uncrosslinked one and more similar to that of the WT protein. If this were not possible (because the fraction of crosslinked channel-like mutant is too small) then the authors could test whether the spectrum of the crosslinked WT protein loses pH sensitivity. This experiment, in addition to supporting the authors' hypothesis of a conformational change in the pore, would also provide a much needed link between the conformational changes observed via the NMR to those prevented by the crosslink.*

The reviewers’ suggestion to measure the NMR spectrum of the cross-linked channel-like mutant is excellent. We attempted to purify the required “Y419only/D417C/E148A/Y445S” mutant, but unfortunately it was prone to aggregation even at concentrations lower than required for NMR experiments. The alternate experiment suggested is to measure the NMR spectrum of the cross-linked WT protein. Given that Y419 is in a loop that is likely quite flexible (as indicated by the fact that cross-linking Y419C does not inhibit function), we are concerned that some conformational change might remain at this position (thereby complicating interpretation of the data). Therefore, we chose to use double electron-electron resonance (DEER) spectroscopy to measure distance at the site of interest (D417C) and provide a link between spectroscopically-detected pH-dependent conformational change and the conformational change prevented by the cross-link.

The inhibition observed upon cross-linking at D417C demonstrates that motions in this region are involved in ClC-ec1’s Cl^-^-transport function. Assuming the pH-dependent conformational change detected by NMR is reporting this same functional motion, then the pH-dependent conformational change involves an increase in the 417-417 inter-subunit distance. We directly tested this prediction using DEER to measure inter-subunit distances on spin labels attached to D417C ClC-ec1 as a function of pH. Consistent with our prediction, the inter-subunit distance increases when the pH is lowered (new Figure 4). DEER measurements on spin-labeled D417C in channel-like ClC-ec1 provide a further link between the spectroscopically detected conformational changes and the cross-linking results: first, a decrease in pH does not shift the intersubunit distance distribution as in the WT background (new Figure 4), consistent with the lack of pH dependence observed in the ^[19]^F NMR experiments on channel-like ClC-ec1 (Figure 3); second, the inter-subunit distance distribution between labels on D417C in channel-like ClC-ec1 at pH 7.5 is greater than that observed in the WT at pH 7.5 and moves towards that observed in the WT at pH 4.5 (new Figure 4). Figure 4—figure supplement 5 shows functional, CW-EPR, and DEER analysis for MTSSL-labeled ClC-ec1 variants.

*4) The authors' finding that the crosslink specifically inhibits the Cl^-^ pathway while leaving the H^+^ pathway largely unaltered is extremely troubling and in stark contrast to their transport model. If H^+^ and Cl^-^ transport are tightly coupled then locking the transporter in one conformational state should result in parallel inhibition of both fluxes. The authors show that this is not the case for the Y445S mutant. The consequence of this observation is that the two processes are not tightly coupled and that H^+^ transport can occur independent of Cl^-^ movement and of any conformational changes occurring in the protein. This finding poses critical issues that need to be resolved:*

*i) As far as we understand, the H^+^ fluxes are uphill transport not efflux. Where does the energy driving H^+^ movement come from if there is no Cl^-^ movement? The only ion out of equilibrium is Cl^-^, all others are at equilibrium. So if there is H^+^ movement it must be coupled to Cl^-^. It could be argued that the crosslink "rescues" the mutation by acting on the intracellular gate, reducing leak or even making the transporter "super-coupled". However this is incompatible with other parts of their arguments. Furthermore, this does not solve the issue of what is the energy source for H^+^ transport extrapolated to the "complete" crosslink conditions and inhibition. Thus, this result is puzzling from a thermodynamic point of view.*

*ii) The stoichiometry of the Y445S mutant is ~39:1, yet the authors see that this mutant mediates ~40% of WT's H^+^ transport rather than ~5% which would be expected for such an altered stoichiometry (assuming that the rate of the mutant is not much different from WT's). This should be clarified.*

*iii) In all proposed models (including the authors') H^+^ transport requires a movement of E148 which, in turn, is coupled to movement of Cl^-^ in the pore. How can this be reconciled with the finding that H^+^ transport can occur with no Cl^-^ movement and with no conformational changes?*

*iv) Protonation of E148 is energetically linked to Cl^-^ binding. How can E148 become protonated/deprotonated if no Cl^-^ is being transported?*

We did not intend to imply that H^+^ transport can occur independently of Cl^-^ movement. Clarification of the Y445S cross-linking result and our interpretation of it is presented here, as well as through additional text in the manuscript.

Y445S is a highly uncoupled mutant that was originally characterized by Walden et al. (2007, J. Gen. Physiol.) to have a Cl^-^/H^+^ stoichiometry of ~39:1. In this manuscript, we find that the D417C/Y445S double mutant is similarly uncoupled, with a stoichiometry of ~43:1, determined from taking the ratio of the Cl^-^ transport rate (850 s^-1^, ~40% that of WT) to the H^+^ transport rate (~20 s^-1^, ~2% that of WT). We determined stoichiometry of this mutant and all CLC-ec1 variants studied here by taking the ratio of the measured transport rates for Cl^-^ and H^+^. This is now clarified in Methods.

Cross-linking the D417C/Y445S double mutant decreases the Cl^-^-transport rate but did not appear to decrease the H^+^-transport rate (original Figure 7; current Figure 6—figure supplement 1). Our original interpretation of this result was that, as the reviewers suggest, the cross-link in effect “rescues” the mutation: as the Cl^-^-transport rate decreases, the stoichiometry increases. This effect does not violate any thermodynamic principles because at the maximal cross-linking (70%), the Cl^-^-turnover rate is ~300 s^-1^, the H^+^-turnover rate is ~20 s^-1^, yielding a stoichiometry of 16:1. This interpretation assumes that the Cl^-^-transport rate extrapolates to a low but non-zero value, such that at 100% cross-linking there is some Cl^-^ transport remaining that is coupled to the H^+^ transport. However, as the reviewers point out, given the uncertainty in measuring such low rates of H^+^ transport (20 s^-1^), we agree that we likely overestimated our ability to conclude whether or not the cross-linking inhibits H^+^ transport in D417C/Y445S ClC-ec1. We have therefore rewritten this part of the manuscript (subsection “Helix P cross-link specifically affects the Cl^-^-permeation pathway”, second paragraph) accordingly.

*5) The authors use CuP to promote crosslinking. However, they find that a large fraction of the thiols becomes oxidized and thus prevented from forming disulfide bonds, which then prevents formation of the crosslink. Have the authors tried other strategies to induce disulfide formation? Maybe ones that do not rely on the generation of local concentrations of reactive oxygen like CuP does? Since the inhibition of the crosslinked D417C mutant is the key piece of evidence linking the observed conformational changes to transport it is important to directly show that a completely crosslinked transporter is indeed fully inhibited.*

We performed cross-linking experiments using HgCl_2_, following the protocol reported by Basilio et al. in their studies of ClC-ec1 cross-linking (Nat. Struct. Mol. Biol., 2014). Unfortunately, HgCl_2_ is a much less effective catalyst of cross-linking the D417C mutant than is CuP (we used the A399C/A432C mutant generated by Basilio et al. as a positive control). Therefore, we have not been able to achieve 100% cross-linking. Whether or not the completely cross-linked transport is fully inhibited is discussed in point 9 below. We remain cautious in our interpretation of the data, and we think that regardless of whether the completely cross-linked transporter is fully inhibited, the results presented are important and informative of the CLC mechanism.

*6) The authors state that "The question remains whether and how these conformational changes are involved in regulating ion binding and translocation during Cl^-^/H^+^ transport." This question is not as open as the authors suggest. Basilio et al. showed that such conformational changes do occur and that they are strictly required for transport.*

Thank you for pointing out that we did not sufficiently elaborate on the important work of Basilio et al. in the Introduction. We clarified the sixth paragraph to emphasize that it is focused specifically on H^+^-dependent conformational change, and we added a more detailed description of the Basilio et al.results in the fourth paragraph.

*7) The authors show (subsection “H^+^-dependent accessibility of Y419 “and Figure 3) that the NMR peak corresponding to Y419 shifts and splits when pH is lowered. The authors interpret this (together with other evidence) as an indication that CLC-ec1 adopts different conformations. Can they use the relative weight of the peaks to estimate what fraction of the time CLC-ec1 spends in the OF state? The concern here is that low pH promotes turnover of the transporter and therefore it is plausible that rather than stabilizing a specific state this maneuver populates multiple states.*

It is certainly possible that multiple states are populated at low pH (and also at high pH). Qualitatively, the relative areas of the two peaks appear similar, suggesting a comparable population between the two pH states. The split in the NMR peak could arise either from two conformational states of ClC-ec1 or from a tyrosine ring flip that occurs slowly on the NMR timescale. Additional experiments would be needed to tease apart these possibilities, however, since the overall interpretation of the data remains the same regardless of the reason for the peak split, these experiments were not prioritized. For example, this could be probed further by a series of temperature experiments to see if the peaks would combine to a single peak at higher temperatures (perhaps supporting the ring-flip), but the inherent instability of the protein at low pH and higher temperature would make this experiment challenging. Alternatively, using a 4F-Phe at position 419 instead of Tyr could also shed light onto the issue, but assessing the functional significance of a Tyr to Phe mutation at position 419, and the additional challenge of generating a Y419F-only mutant, were beyond the scope of our efforts. Our goal in these experiments is simply to shift the conformational equilibrium to increase the population of the OF state. As described above, we now show H^+^-dependent conformational change detected by DEER experiments on MTSSL-labeled D417C CLC-ec1 (new Figure 4). In the WT background, the distance distribution of the label at 417C is clearly pH-dependent. Although in principle DEER can be used to quantify populations, the range of distances present precludes an accurate quantification of the populations. However, the major point – that there is a pH-dependent shift in conformational equilibrium – is confirmed.

*It is hard to follow the authors' reasoning when they conclude (subsection “H^+^-independent accessibility of Y419 in channel-likeClC-ec1”) that "In channel-like E148A/Y445S, the accessibility of Y419 to TEMPOL indicates that the external Cl^-^-permeation pathway of the channel-like ClC-ec1 variant adopts a conformation in solution different from that observed in the crystal structure and similar to the conformation adopted by WT at low pH." Y419 is located quite far from the ion transport pathway, so while their data indicates that the environment surrounding this residue is indeed different in the WT and channel-like backgrounds, we do not see how that the same argument can be readily extended to the Cl^-^ pathway.*

We agree this was an overstatement at this point and have modified the text accordingly: “In channel-like E148A/Y445S, the accessibility of Y419 to TEMPOL indicates that the channel-like ClC-ec1 variant adopts a conformation in solution different from that observed in the crystal structure and similar to the conformation adopted by WT at low pH.”

*8) A point omitted here which seems might go to the heart of the issue is that the PDB includes a single CLC crystal structure at pH 4.6, where the alternative outward-occluded structure of this manuscript is proposed to exist* –

*the original 3.0 A structure of CLC from Salmonella (PDB 1KPL). We wonder if you see any differences in helices P, N, or F in this 'Glu_ex_-down' that might differ from the high-pH E. coli CLC structures? Any hints here?*

We thank the reviewers for the comment. Indeed, one would expect to see structural differences in a CLC crystallized at low pH, though of course a crystal-seeding step need not involve the predominant species in solution, and thus the Le Chatelier effect can lead to crystallization of a minority species. Nevertheless, one would hope to catch a glimpse of the OF_occluded_ or OF_open_ states. Instead, the high-pH (1OTS) and low-pH (1KPL) X-ray structures are virtually identical (C-α RMSD 1.0 Å; Helix N/P all-atom RMSD 0.7 Å). In addition, the side chains of key residues (Glu_ex_ (E148), F357, L361, L411, M415, D417, and Y419) are similarly positioned. The radius profile of the Cl^-^ transport tunnel reveals an extracellular-gate constriction as observed in ClC-ec1 (plus, interestingly, an additional constriction region towards the extracellular side of the ion-permeation pathway). This information has been added to the manuscript in Figure 1—figure supplement 1 and Figure 9—figure supplement 1.

*9) We think that the authors are underinterpreting the crucial Figure 4 data* – *the correlation of fraction of protein crosslinked and fractional inhibition of transport. They ask in the Discussion (fourth paragraph): why doesn't crosslinking completely inhibit activity?* – *and then they go into some handwaving to suggest various possibilities. But in Figure*

*4C and D it could be concluded that to a first approximation, crosslinking fully inhibits. Granted, you cannot achieve 100% crosslinking because of the harsh CuP conditions, which, as you show, produce higher oxidation states of the thiol; but the linear correlation drawn on the figure suggests that if you could have gotten to 100%, you'd have less than 20% activity left, which might not be distinguishable, given the assays, from zero. It's of course always virtuous to be conservative in data interpretation, but this possibility ought to be mentioned along with the rather anodyne alternatives offered up to explain the 'residual activity' of the crosslinked protein.*

We thank the reviewers for making this point. To address the issue, we re-plotted all of the cross-linking data as individual points (rather than averages of 2-4 experiments) indicating absolute (instead of relative) turnover rates, fit these data to linear regressions, and estimated the 95% confidence interval for the activity at 100% cross-linking. These values are summarized in the new Table 1. It does appear that there is some residual activity at 100% cross-linking, at least in some of the mutants, though given compounding uncertainties in the experimental measurement it is possible that the residual activity is not distinguishable from zero. For example, the turnover rate of un-crosslinked D417C/E148A is as low as the extrapolated value for the turnover of fully cross-linked D417C (~270 s^-1^), yet cross-linking of D417C/E148A further reduces this activity (Figure 6, Table 1). This result emphasizes that the extrapolated values provide only an estimate for turnover of fully cross-linked transporters. We have now included in Discussion (fourth paragraph) the possibility that the residual activity is not distinguishable from zero.

*10) More on Figure*

*4D. You rather do yourselves a disservice by presenting the flux data as you do. Your point is to illustrate the effect of crosslinking on the initial rate of transport – i.e. at early times. But the bulk of the figure is taken up, purposelessly, by the big jump by detergent that gives you the normalization value. we would leave out that big jump, expand the y-axis to put the early-time flux traces on a full-scale display, and just tell us in the legend or Methods, that fluxes are normalize to values obtained after dissolving the liposomes with Triton. This is not merely a matter of display-aesthetics; the reader would benefit from actually being able to see the degree of inhibition, which is hard to perceive with the meaningful parts of the traces all compressed into the lower 20% of the figure.*

The figures have been revised as suggested.

11) Figure

*4F. Why do the efflux traces level off at only half the liposomes at the protein density you are using? I would expect all the liposomes to contain active CLCs.*

For experiments on channel-like CLC, we used the low range of protein density reported in Methods, 0.2 µg per mg lipid. At this density, we expect roughly half of the liposomes to contain active CLCs, as has been observed by others (for example Figure 6 of Walden et al., 2007). We have added a sentence to the Methods to make it clear that the low end of the range was used in experiments on channel-like ClC-ec1.

12) In the MD simulations it would be good to know a bit more about how the special motion-types were identified. Specifically, the word "extrapolated" (subsection “Potential gate-opening motions inWT and cross-linkedClC-ec1”) in finding the looked-for collective motions raises skepticism about what, exactly, those Δ

-r distances are in Figure 8. Do they represent actual frames in which the pathway is seen to be 2 A wider, or some sort of widening imagined by some sort of sophisticated extrapolation algorithm?

Thank you for requesting this important clarification. The idea behind our analysis is that a simple equilibrium simulation may not be long enough to show a complete collective motion (due to timescale limitation), *but* one may capture the beginning of it. So, we first use PCA-based collective motion analysis to identify motions from an equilibrium simulation, then we induce larger conformational changes along those motions (similar to the idea of exciting particular normal modes). For example, suppose a 0.5 Å separation of two domains is observed as a collective motion during an equilibrium simulation. We next excite this mode (i.e., move the protein further along this particular deformation/vibration/mode), finding the displacement vector that defines this mode and multiplying it by a factor larger than 1 so as to deform the protein along the identified pathway.

We do this analysis for the top 20 collective motions of both WT and cross-linked species; how far we go along the vector is based on an arbitrary RMSD that we define as a constant for all of the collective motions that we studied individually. The point of the analysis is to find out how many of the top 20 collective motions of the protein that have been detected during the equilibrium simulation can result in significant motions in the gating regions of the protein. The analysis (Figure 8) shows that in the cross-linked species, the coupling between the global fluctuations of the protein (top 20 collective motions capturing 75% of the motion observed in our equilibrium simulations) and gating elements has been reduced. In other words, the protein dynamics “energy” is less funneled into motions that can be used for gating. Or put another way, thermal fluctuations of the protein are now along other degrees of freedom and less engaging gating elements. We have extensively revised the Results and Methods sections (subsections “Potential gate-opening motions in WT and cross-linked ClC-ec1” and Analysis of collective motions of protein”) to clarify these points.

*13) We would take issue with the cavalier use of the term "large-scale motions" to describe the sorts of wiggling you are envisioning. Of course, use of this phrase is subjective, but we think that most structural biologists would refer to these as 'breathing motions.' To call them large-scale might be viewed by many as tendentious.*

The use of the term “large-scale motions” has been eliminated as suggested.